# Spiritual Reports from Long-Term HIV Survivors: Reclaiming Meaning While Confronting Mortality

**Kyle Desrosiers** [1,2]

1   Conflict Resolution and Mediation, Department of Social Sciences, Tel Aviv University,
    Tel Aviv 6129302, Israel; kyled@mail.tau.ac.il or desrosie.kyle@gmail.com
2   University Scholars Program, Baylor University, Waco, TX 76798, USA

**Abstract:** *Reports from Long-term HIV Survivors: Reclaiming Meaning while Confronting Mortality* presents research completed by Kyle Desrosiers in conjunction with the Baylor University Institute for Oral History. Applying lifespan theory to spiritual development, it discusses the narratives of four American long-term HIV survivors from Latter-day Saints, Roman Catholic (2), and Conservative Jewish backgrounds. The fifth profile is from a Protestant pastor with an HIV ministry in a rural area. These profiles are five selected from among 10 interviews with HIV-positive people and caregivers across America now archived by the author at Baylor University. Questions directing this research were: how does HIV status affect participants' relationship to their religious communities, identities, and spiritualties?; what narratives emerge from lifespan perspectives of HIV-positive and queer participants?; and what spiritual practices, mythos, and beliefs evolve/remain as a product of living at the margins of religion and society, alongside coping with a deadly global epidemic? This project reports narratives of change, continuity, and meaning-making to discuss how several gay/queer men from a range of ethnic and faith backgrounds have used spirituality and worldview to navigate life.

**Keywords:** HIV/AIDS; health; healthcare; queer theology; lifespan theory; oral history; religion; epidemic; LGBTQ studies

---

## 1. Introduction: Lifespan Spiritual Narratives of HIV-Positive Men

Have you ever watched your best friend die (what for)

Have you ever watched a grown man cry (what for)

Some say that life isn't fair (what for)

I say that people just don't care (what for)

They'd rather turn the other way (what for)

And wait for this thing to go away (what for)

Why do we have to pretend (what for)

Someday I pray it will end

—*In this Life*, Madonna 1992[1]

I will say to God, my Rock, 'Why have You forgotten me?

Why should I walk in gloom under the oppression of the enemy?'

---

1   (Madonna 1992).

> With murder in my bones, my oppressors have reproached me by
>
> saying to me all day long, 'Where is your God?'
>
> <div align="right">—The 42nd Psalm, v. 10–11.[2]</div>

Few phenomena reveal the character of a society like its response to tragedy. This is especially evident in the ongoing story of HIV in America. Since the 1980s, the responses of religious communities, people of faith, politicians, and those who stood to profit from the medical-pharmaceutical industrial complex have needfully endured great scrutiny and criticism. Each human being whose life has been directly or indirectly touched by the HIV virus and its comorbidities, had to evaluate their experiences from religious, philosophical, or meaning-seeking lenses. Humans are religious animals—theists and atheists alike—and it is natural that those living at the margins have unique relationships to various structures and institutions of belief. The HIV/AIDS epidemic—as many historians have argued[3]—represents a tragedy for the great apathy and ambivalence it was faced with by the dominant American society and government. Likewise, it also represents a history to be studied, in which society members can be divided into groups: those directly affected, those committing antagonism, those complicit with antagonism, or those allies who emerged heroic. As Charles E. Rosenberg observes of the study of epidemiology, "the unique historical moment of an epidemic allows the historian to study the effects of a randomly occurring stimulus, against which the varying reactions of [a population] can be judged."[4]

Indeed, because HIV/AIDS primarily affected gay and queer men, but also often trans women, sex workers, intravenous drug users, and poor communities frequently made up of Black, Brown, and Indigenous people, the dominant cultural hegemony deployed religious and "moral" responses, ripe for exploitation for political gain or status quo maintenance.[5] According to historians and activists, the crisis revealed the true ethical colors of those whom authority and some common consent had given a platform—clergy people, preachers of all sorts, the Ronald Reagan administration, and various lobbyists of pharmaceutical companies, or causes such as anticontraception and antigay rights.[6]

In pop culture—such as in recent television programs like Ryan Murphy and Janet Mock's *Pose*[7], recent novel *The Great Believers* by Rebecca Makkai[8], or older queer works such as the creative nonfiction work of Randy Schilts and the plays and advocacy of ACT UP founder Larry Kramer and the works of Tony Kushner[9]—the struggle among many religious institution and religious "moral voices" against HIV-positive and queer bodies may appear to the audience as a totally antagonistic relationship. This, however, is not the intent of most artists, and such an understanding neglects the personal spiritual narratives that HIV-positive persons developed out of necessity to answer questions of meaning-making. In addition to the trauma and violence many participants reported being caused by doctrines and individuals, many also reported how specific religious values, practices, and experiences forged positive spiritual meaning and deepening contemplation. Religion is a great source of challenge for HIV survivors. Religious beliefs and institutions may create situations that are simultaneously both helpful and hurtful, and marginalized people may be compelled to work hard to winnow the wheat from the chaff. This article makes no generalized claims about HIV/AIDS experience, but rather

---

[2]  (Chabad 2020). Adapted from the Hebrew text of Chabad.org online database.
[3]  See the following works on religion and AIDS/HIV and culture/history: (Schilts 1987; McNeill 1998; Slack 1996; Cleworth 2012; Bluthenthal et al. 2012; Derose et al. 2010; Pierce and VanDeVeer 1988; Haddad 2011; Trentaz 2012; Björklund and Larsson 2018; Brier 2018; Epstein 1996; Green 2016; Pepin 2011; Petro 2015; Silverman et al. 2020).
[4]  (Rosenberg 1992).
[5]  (Shomanah and Kanyoro 2005; Monette 1998; Corea 1993).
[6]  See the following for information on the Reagan administration and pharmaceutical corporations and the HIV/AIDS crisis: (Cleworth 2012; Haddad 2011; Rofes 2012; Pierce and VanDeVeer 1988; Green 2016; Saleebey 2001; Murphy 1994).
[7]  (Murphy and Mock 2018–2019).
[8]  (Makkai 2018).
[9]  See: *And the Band Played On* (Schilts 1987); *Reports from the Holocaust: The Story of an AIDS Activist* and *The Normal Heart*, (Kramer 1985); and *Angels in America: A Gay Fantasia on National Themes: Revised and Complete Edition*, (Kushner 2013).

centers on the narratives directly from long-term survivors, of diverse ages and ethnicities. The focus among this corpus is lifespan stories of spirituality change and continuity.

This author draws upon the medical-socio-narratological work of Arthur Frank[10], whose words reveal the significance of storytelling in the struggle to bring to light instances of injustice and marginalization that those with power would rather forget. Indeed, the stories of long-term HIV survivors join the throng of raised voices among all those who must also cope with their mortality and the "value" that society does or does not bestow upon their bodily and spiritual integrity. LGBTQ persons, Black, Brown, and Indigenous people, other communities of Color, those living with disabilities, and those who are uneducated or live in poverty, are communities at the margins. Many among these communities stand as exemplars of what it means to resist oppression by authoritative, exploitative religious doctrinal interpretations or hierarchy to develop their own narratives for the sake of personal and collective human flourishing.[11] The pursuit of human flourishing—including the right to dignity, autonomy, health, meaningful work, and community bonds—is a religious concept found in many faiths, such as Christ's offer of abundant life in the Gospel.[12] Together these stories of life and death at the margins unite, and call out in the wilderness hoping to bring about a future of better possibility. As Frank writes:

> How might people's lives change if they heard their own stories with enhanced reflective awareness and if they heard others' stories with a more generous sense of what makes these stories more viable representations of the lives those storytellers live?... Stories are representations not so much of life as it is, but of life as it is imagined, with that imagination shaped by previous stories. Storytelling is a dialogue of imaginations. This dialogue is real in its consequences for how people act.[13]

Scholarship suggests that cooperate religious-ethical systems are closely linked to individuals' spiritual health outcomes. A 2006 study showed links between "healthy" and "unhealthy" spirituality and the quality of life and physical and emotional well-being for people living with HIV/AIDS.[14] This research suggests that there is a direct link between unhealthy spirituality—e.g., harmful doctrines or abusive relationships with clergy—and poorer immune system function among persons living with chronic illness. According to this study and other medical researchers[15], poorer health status is statistically associated with less social support, poorer spiritual well-being, and depression. Communal and doctrinal/dogmatic components of faith systems affected health outcomes positively or negatively, whereas denominational affiliation or personal level of belief did not result in significant distinctions among the sample. Ultimately, community and a strong sense of positive meaning were the most significant factors. Healthy religious community and affirming religious doctrine were two components that were largely absent for the generation of gay men who endured the worst of the American HIV/AIDS Crisis, who were by-and-large neglected and antagonized. They were told that not only did they not belong in life—but could also face total separation from God in death.

---

[10]  (Frank 2011).

[11]  Coming out of Latinx and Black liberation theology, feminist theology, and gay-lesbian liberation theologies, the late twentieth century discipline of queer theology has explored ways in which LGBTQ people, and other groups marginalized on the basis of sex and gender, find meaning, and purpose, and a true connection to faith. It rejects the teachings of "otherness" and "intrinsic disordered-ness" that some theologies and doctrines teach about queer people and human sexuality. Instead, it critically views doctrine and scripture as contextual; it is critical of authoritative works as being written by those in power to maintain the status quo. Queer liberation theology strives to reclaim the doctrines and practices that have been used to oppress and exploit, and instead reinterpret them in context. Queer/LGBTQ people, but also women and racial and ethnic "others" have been able to reclaim autonomy over the intrinsic goodness of their faith–whether in scriptures, traditions, practices, or their own spiritual understanding. See: (Shallenberger 1998; Guest 2006; Shore-Goss 2013; Albert et al. 2001; Tonstad 2018; Cheng 2011; Boisvert and Johnson 2012; Knauss and Mendoza-Álvarez 2019).

[12]  (The Gospel according to John, 10:10).

[13]  (Frank 2011, p. 50).

[14]  (Yi et al. 2006).

[15]  (Cotton et al. 2006; Doolittle et al. 2018; Litwinczuk and Groh 2007; Siegel and Lekas 2002; Grill et al. 2020).

Both communal belonging and private spiritual or meaning-making contexts are deeply important for the emotional and physical well-being of persons living with chronic illness and stigma, such as is the case of many long-term HIV survivors.

In addition to Frank's theory on the practicing of dialogical narrative analysis, this research was also completed keeping in mind the framework of lifespan theory, which has been pioneered and utilized by many in psychology, the helping professions, and care providers for those living with mental, emotional, and physical disabilities or conditions.[16] Religious views (including nonreligion, atheism, etc.) are integral to one's being, and interact with all facets of a human life. Lifespan theory honors the complexity of the change and continuity of beliefs and conditions of human lives. The theory sees its origins from the psychosexual development work of Sigmund Freud[17], the psychosocial theory work of Erik Erickson[18], Jean Piaget's cognitive theory of development[19], and contemporary psychological models of human development over time.[20] Erickson's theory on the identity and personality are especially useful for understanding how beliefs and relationships change over time, which this author understands to be a process inseparable from spirituality, religion, and/or cosmology.

The study of religion's relationship to human rights frequently lacks comprehensive coverage of personal spirituality. Discussing religion—particularly when evaluating "spiritual journeys" as this project does—demands an investigation that goes beyond a detached objectivity. Religion—whether it is Sunni Islam, Mormonism, Episcopalianism, or atheism—cannot be studied as though it were purely academic. Spirituality is a mechanism by which individuals unite their internal self with the outside world, respond to stimulus, and strive to comprehend perennial questions of meaning. The religious imagination is deeply ingrained in all people, even those whose religion is only grounded in material observation and scientific theory. Thus, bearing this in mind, this project centers individual humans and their particular relationship with spirituality. It is important to approach religion and spirituality with attention to pluralism; that is, valuing and affording dignity to every unique worldview. At the same time, respecting the dignity and autonomy of human beings means representing their definitions of religion as intrinsically sacred or meaningful to the individuals who possess them. This unique perspective will hopefully contribute to the existing literature and provide insight on the relationship between HIV/AIDS and spirituality.

## 2. Lifespan Narratives of Belief, Sexuality, and Health: Presenting Findings

In this article, five participants have been selected from an archive of interviews the author conducted with Americans from an array of demographics. Perspectives profiled here include: Black Roman Catholic; Latinx Roman Catholic; white Latter-day Saints; white Conservative Jewish; and white mainline Protestant—though some participants reported no longer identifying with the faiths into which they were baptized or initiated, instead finding less dogmatic or spiritual-not-religious paths. Questions directing this research were: how does HIV status affect participants' relationship to religious community, identity, and spirituality?; what narratives emerge from lifespan perspectives of HIV-positive and queer participants?; and what spiritual practices, mythos, and beliefs evolve as product of living at the margins of religion and society, alongside coping with a deadly global epidemic? Particularly, data collected from these research questions were organized and synthesized to display narratives regarding the following themes: family, culture, and religious affiliation overtime; religion/spirituality and sexuality; personal and collective HIV and/or religious trauma; dealing with loss and reconciliation; and hopes and anxieties about the future of HIV/AIDS, the LGBTQ community,

---

[16]  (Berger 2017;  Broderick and Blewitt 2010;  Carey 2003;  Doenges et al. 2014;  Giardino et al. 2003;  Platt et al. 2015; Rossi and Association 1985).

[17]  (Freud 1949).

[18]  (Erickson and Erikson 1997).

[19]  (Piaget and Donald 1977).

[20]  (Baltes et al. 2007).

and the world. These all represent commonalities, as well as points of difference in how various participants responded to certain stimuli or conditions. They are not generalizations of all individuals living with HIV, but rather a synthesis of the stories some individuals living with HIV have shared.[21]

### 3. The Beginnings: Childhood Formation within Family and Faith Institutions

Each participant experienced a different introduction to the life of faith or religion. This study began by investigating experiences relating to religion, ethnicity, identity, and family as important components of childhood development—a very significant psychological period identified in lifespan theory.[22] Most participants were queer or gay men, and so in childhood—long before learning how to cope with their HIV statuses—already knew the struggle of "difference".

One participant, Michael, a 59-year-old white Jewish gay man from Los Angeles, California[23], explained feeling different from others in his hometown community in small-town Pennsylvania:

> I felt like an outsider for a number of reasons. Being Jewish was only one of them. Every time there was a Jewish holiday, for example, when I was in elementary school, me and the other one or two Jewish kids would be asked to say a couple of words about, 'What is Hanukkah,' and, 'Bring in the menorah for show-and-tell,' and that sort of thing, but I always felt like they almost laughed at it. I didn't feel like it was something that was appreciated as legitimate. But mostly, I felt different because I didn't know what it was at the time, exactly. At about 12 years old, I figured out that I was gay and that most of my feeling different had to do with not being interested in girls, not being interested in sports, and being Jewish was only a piece of the puzzle.[24]

Identity-related childhood formation also shaped the way that another participant, Ron, a 35-year-old Black gay man from Dallas, Texas[25], viewed his relationship to worldview, belonging, and understanding marginalization, beginning with childhood:

> So, I actually grew up in Houston and we were pretty poor. I say 'pretty poor'—we were really poor, but I was raised by my grandmother and my grandmother couldn't read or write, so she couldn't teach me those basic skills, but she taught me right and wrong... and one of the things that was right for her was religion.[26]

Ron explained that he was a precocious child, intelligent enough to enter first grade at age three. He says that this was one of the first indications that he was different from other people. He says that this difference instilled a sense of shame. It was this precociousness and natural tendency to question things that led to a complicated relationship with Roman Catholicism:

> It's like, 'Okay. You're going to be baptized here, and you're going to do this, and you're going to be an altar boy.' I always felt pressured to be something that I didn't know how to be because I only knew how to be me, and I didn't realize that I was different. All I knew was that society was telling me that, 'Boys are supposed to do this, girls are supposed to do this,' and I just didn't know that I wasn't doing the right things.[27]

---

[21]　Preserved at the Baylor Institute for Oral History, Baylor University, Waco, TX. Please see the Supplementary Materials on pg. 33.

[22]　(Erickson and Erikson 1997).

[23]　Michael Sugar, interview by Kyle Desrosiers, October 2, 2019, telephone interview between Los Angeles, CA and Waco, TX, transcript, Baylor University Institute for Oral History, Waco, TX.

[24]　Michael Sugar, interview by Kyle Desrosiers, October 2, 2019, telephone interview between Los Angeles, CA and Waco, TX, transcript, Baylor University Institute for Oral History, Waco, TX. Interview transcript page 4.

[25]　Ron Wilson, interview by Kyle Desrosiers, September 19, 2019, in Dallas, Texas, transcript, Baylor University Institute for Oral History, Waco.

[26]　Ron Wilson, interview by Kyle Desrosiers, September 19, 2019, in Dallas, Texas, transcript, Baylor University Institute for Oral History, Waco

[27]　Ron Wilson, interview by Kyle Desrosiers, September 19, 2019, in Dallas, Texas, transcript, Baylor University Institute for Oral History, Waco

Another participant, Jesús, 59-year-old gay Mexican-American man from Los Angeles, who was raised Roman Catholic[28], associates Catholic rigidity with harmful cultural understandings of gender and sexuality. Jesús was born and raised in central Mexico, and explains the methods he cultivated to strengthen himself against harmful messages and abusive norms:

> I lived my life without having the turmoil about me being gay—because when you're surrounded by, 'You're going to hell if you do this,' people get traumatized ... [but] I didn't have that turmoil even if I was going to church when I was growing up. It was almost like that safety of knowing I have my own personal connection with God, and I know I'm a good person, and I'm not hurting anybody, and whoever I am, this is what I am.[29]

Jesús was nonetheless conscious of the conditions of life that are defined by Roman Catholic Christianity but also Mexican-Latinx cultural norms of the mid twentieth century, especially how they related to gender roles and homosexuality. He also recognizes the positive role that faith played in shaping people are loving and compassionate:

> Many of my uncles and people are like the stereotypes of the macho Mexican person that follows Christianity and Catholicism a lot and everything else is bad or wrong,... [but] the person that I adore most in my life, my grandmother—she was the best example of simply doing good or loving someone, and she was a very Catholic woman, but she was the most accepting woman of things. She was very Catholic with the rosary by her bed all the time, and I'm sure she was praying for me every day.[30]

Robb, a 56-year-old white gay man from Dallas, TX, became a baptized Latter-day Saint when his family converted to it while he was young. A child in a family of converts, he discusses the way in which his tradition discouraged dissention.[31] Similar to Ron, he credits his intelligence as a child and a natural proclivity toward questioning for inspiring him to challenge what he could not understand:

> If you've been in the Mormon Church, they do home visits weekly, and you have three hours of church plus your preparation for whatever your role [outside of Sunday church meetings, i.e., seminary, missions, etc.] ... I mean, really, you do not get any time alone to even clear your head outside of the dogma that you're being barraged with.[32]

Robb explains that the most challenging part of being a gay Mormon meant negotiating the Church's rigid gender and sexuality structure, a situation that many of the participants from other faith traditions also experienced:

> My self-esteem was almost destroyed, I thought about suicide going through that reconditioning [LDS gay conversion therapy], how friends of mine have had these dark nights of the soul with their religion, being told they were these awful things—I don't know why we [the church authorities] scare them so much. Part of what I think is this: to be a real dominator society, you have to be real machismo. I think we [gays] shine a light on what they can't—that machismo is a weakness, that it's too hard. It's too rigid, it's close-minded, and it needs to be more flexible.[33]

---

28  Jesús Guillen, interview by Kyle Desrosiers, December 20, 2019, via telephone call between San Francisco, CA and Waco, TX, transcript, Baylor University Institute for Oral History, Waco.

29  Jesús Guillen, interview by Kyle Desrosiers, December 20, 2019, via telephone call between San Francisco, CA and Waco, TX, transcript, Baylor University Institute for Oral History, Waco. Interview transcript page 6–7.

30  Jesús Guillen, interview by Kyle Desrosiers, December 20, 2019, via telephone call between San Francisco, CA and Waco, TX, transcript, Baylor University Institute for Oral History, Waco.

31  Robb Ivey, interview by Kyle Desrosiers, September 20, 2019, Dallas, TX, transcript, Baylor University Institute for Oral History, Waco.

32  Robb Ivey, interview by Kyle Desrosiers, September 20, 2019, Dallas, TX, transcript, Baylor University Institute for Oral History, Waco

33  Robb Ivey, interview by Kyle Desrosiers, September 20, 2019, Dallas, TX, transcript, Baylor University Institute for Oral History, Waco. Interview transcript page 27.

Yet, as participants explained, the HIV virus would indeed come to put significant limits on their lives. If parents, clergy, or culture had told them anything was possible as children, many of them would later hear from the same communities that they were no longer welcome or broken beyond repair. For many, finding a way to glean goodness from within traumatic childhood faith and cultural life proved difficult, but others were able to find ways to reclaim practices in ways that were healthy and beneficial. This will be explained more in later sections.

## 4. Understanding Sexuality: Discovery, Liberation, and Trauma

Each participant shared stories of their development of self over time through adolescence and young adulthood, as they were gradually able to separate from their families and communities and investigate their identities for themselves. The included experiences of coming out and coming into self.

One, participant, Michael, told about the time he came out to his parents, who were Holocaust survivors from Europe, and once lived under the norm of persecution and genocide. Because of—or in spite of—their own background, they wanted their son to assimilate and lead what they viewed as a "normal" life. It took them years to understand Michael's sexuality as "normal", and not view him as flawed or sick:

> When I came out to my parents, my mother said, 'Anybody that's sick and is not willing to get help is not a part of this family,' and that was very hurtful. We didn't talk for a long time. And even after my HIV diagnosis, I didn't disclose my diagnosis to my parents. Even though we were speaking again by that time, I didn't disclose my diagnosis to my parents for 10 years... I told them, and the way that I said it was that I had HIV and, 'I absolutely need to know if you're going to be there for me." And at that time, I had a boyfriend and I said, 'Don is my partner. He is family, and if anything should happen to me if I were to get sick, I would certainly want you to be there, but if you can't accept Don as my family, you would not be welcome.' It took them a minute to digest this new piece of information, but they came around and really latched onto that unconditional love that really broke through whatever other feelings they might have had—and I think, 'Wow.' If I hadn't disclosed that to them, how much love I would've cheated myself out of.[34]

Michael's parents were eventually able to develop an ethic of solidarity with their son who was persecuted for being different, in spite of their Eastern European cultural norms:

> I think that my parents' experience with the Holocaust maybe doubly instilled in them that idea about how careful you have to be, and what things are okay, and the price that you pay for being anything other than what's right, and so, they didn't ever really want to know all the details about my test results and what I was experiencing; they just wanted me to be well.[35]

Similarly, Robb, explained that for him it was also because of the trappings of his tradition—and not totally in spite of it—that he was able to find liberation. He began his journey of discovery from within an orthodox theological framework while he was abroad with the Church of Jesus Christ of Latter-day Saints on a mission to evangelize in Japan—a commitment Church members in good-standing are expected to make:

> I thought, 'I'm gonna be obedient, and maybe God will help me.' Here I am on my mission. I start to get a crush on my companion, who was a Japanese Elder[36]. He was so sweet.

---

34  Michael Sugar, interview by Kyle Desrosiers, October 2, 2019, telephone interview between Los Angeles, CA and Waco, TX, transcript, Baylor University Institute for Oral History, Waco, TX. Interview transcript page 26.
35  Michael Sugar, interview by Kyle Desrosiers, October 2, 2019, telephone interview between Los Angeles, CA and Waco, TX, transcript, Baylor University Institute for Oral History, Waco, TX. Interview transcript page 27.
36  A title for men in the Latter-day Saints Church polity.

The Japanese people are beautiful. About halfway through my mission, I lost my urge to push my message on others. I was getting more curious about them, observing how harmonious they are with nature, how considerate they are. I thought, 'I could really pause a moment to understand them better before telling them what they need to know.'[37]

Robb explains that his experiences with beauty, kindness, and new and different philosophies while in Japan inspired him to think more broadly and find goodness and beauty where he had previously been told he could not. When he came back, he was still "going through the motions" and attending Brigham Young University:

I'm going to these sermons and stuff and they're telling me, 'Now, your next mission is to find a fine young daughter of Zion and get married and fill the earth with your spawn—your children. That's your next mission.' (laughs) 'Okay, but I'm not sure I'm really the one to do this.' So, I submit myself to the reconditioning [conversion therapy], and it just destroys your self-esteem and just makes you numb.[38]

For Michael, who self-describes as a lover of film and pop culture, it was only appropriate that the first positive (and indeed, only) media messages he saw about homosexuality was through the spiritual act of watching television, in an era when homosexual lifestyles and gay rights were only just beginning to enter mainstream discourse—and very seldom portrayed in a positive light. He was also 12-years-old, and it was 1972:

It was a film on TV called 'That Certain Summer' with Hal Holbrook, and Martin Sheen and Hope Lange, and it was about a gay couple living in San Francisco and one of them had previously been married and had a child, and the child comes to visit them, and they try to hide the fact that they're gay, but the child somehow finds out, and gets scared and runs away, and they have to scour the city to find this little boy, and then, to bring him home and to explain to him that his father is gay, and what this all means, and that this is about love and this isn't something to fear or to hate. It was not only the first real—the first time that there were real gay characters on a television movie, but the first time that gay characters were also portrayed in a positive light, and sitting there on the sofa with my parents in the living room in York, Pennsylvania, I realized that that's who I am. But, I also learned that it's something that I'm supposed to keep secret until I can move to the big city.[39]

## 5. The Onset of HIV: Personal and Collective Memories

The early 1980s represented a world-altering turning point for the older generation born around the 1950s–1960s, who had come of age as gay in the time of sexual liberation and the early gay rights movements—a time free from AIDS. This generation had only recently begun to discover the delight of freedom and community—when the mysterious virus began to take the lives of healthy, young gay men, starting on coastal cities and spreading eventually in every gay community, from Chicago to Seattle to Fort Lauderdale to Galveston.

Michael discusses the early 1980s Los Angeles, CA when HIV first came on "the scene," and he began to see friends and strangers become very sick and die painful deaths from illnesses that were not supposed to take young men. It came slowly, then in the fashion of epidemics, increased exponentially, affecting a critical level of the LGBTQ population. For Michael, and for other gay men participants,

---

[37] Robb Ivey, interview by Kyle Desrosiers, September 20, 2019, Dallas, TX, transcript, Baylor University Institute for Oral History, Waco. Interview transcript page 8–9.

[38] Robb Ivey, interview by Kyle Desrosiers, September 20, 2019, Dallas, TX, transcript, Baylor University Institute for Oral History, Waco.

[39] Michael Sugar, interview by Kyle Desrosiers, October 2, 2019, telephone interview between Los Angeles, CA and Waco, TX, transcript, Baylor University Institute for Oral History, Waco, TX. Interview transcript page 4.

the growing epidemic represented a tremendous loss of innocence after a few decades of increasing sexual liberation and improved gay rights:

> I lived in West Hollywood, where there were huge numbers of gay people. I felt safe and sort of at home, and it really changed me... and then AIDS came along. I remember, in 1981, I used to go with a bunch of friends. We would work out together at a gym in West Hollywood, and then afterwards we'd all go to this local coffee shop and we'd eat. One night, somebody at that table—there was a group of eight of us that you see, and one guy that was at the table asked if anybody had seen the article in the *New York Times* about what was called the gay cancer. We talked about it and we all dismissed it as, 'That couldn't be. That's impossible.' It wasn't plausible. That was in 1981, and I'm the only person, by the way, that was at that table that night who's still alive.[40]

As Michael, the gay community, and their allies discovered, the mysterious "gay cancer" was indeed real. As more and more people got sick and died, information, service, and governmental aid was slow to follow the epidemic flood:

> I read about it in the newspaper, I just remember that as the first time that it appeared on my radar, and then, slowly, you'd start to hear about people getting sick and dying in New York and in San Francisco. It seemed, at least, in my experience, that it happened in those places first, and then, all of a sudden, the first person I knew was sick. His name was Adam... I remember when we had to get into the hospital and we called the ambulance. As soon as the ambulance people knew that he had AIDS, they wouldn't take him. They left him on the pavement, and so, we had to get him to the hospital another way and we really had to fight for him to get care—for people to even touch him, rather than just leave him in the hallway. At that time, when you visited an AIDS patient, there would be instructions on the door about how to visit an AIDS patient, and you'd have to put on a gown, and a mask, and gloves, and in some instances, that's how people—that's sometimes how family members found out that their son, their child, or their sibling had AIDS, because they arrived at the hospital and found those instructions on the door. And then, after Adam, soon, it was another one and another one and another one. That would've been, like, 1984, and then I got my own diagnosis in 1985.[41]

Though Ron tested positive in the mid 2000s, he still experienced feelings of fear, apathy, and the fear of imminent death, though that would be less and less the scientific reality at this point in history, as medicine improved. Perhaps this is also why Ron has since chosen the path he is on, HIV/AIDS nonprofit advocacy. He remembers the difficult first few months understanding his positive diagnosis:

> I felt the responsibility to tell this person that I was dating, 'Hey, I got a positive diagnosis,' but I had to do it quickly. I couldn't just sit on that. I was like, 'Oh, my God. What if I give it to him?' That was my thought, and so, I told him after processing that, and he sort of cut me off and I just thought of myself as shit. That was probably the other reason why I felt like, 'Okay. I can just go die now. It's fine,' because I'm 23 and I just affected someone else's life, in my head.[42]

---

[40] Michael Sugar, interview by Kyle Desrosiers, October 2, 2019, telephone interview between Los Angeles, CA and Waco, TX, transcript, Baylor University Institute for Oral History, Waco, TX.

[41] Michael Sugar, interview by Kyle Desrosiers, October 2, 2019, telephone interview between Los Angeles, CA and Waco, TX, transcript, Baylor University Institute for Oral History, Waco, TX.

[42] Ron Wilson, interview by Kyle Desrosiers, September 19, 2019, in Dallas, Texas, transcript, Baylor University Institute for Oral History, Waco. From interview transcript page 12.

Jesús remembers feeling in denial of his diagnosis. Indeed, people who are diagnosed as HIV-positive—as others who receive terminal diagnoses sometimes do—often go through the stages of grief similar to those who have lost a loved-one[43]:

> In those days there were many false positives and many false negatives, so I tried to do that test afterwards, like, four more times thinking it would come back negative... But no. It was positive, and that's another part that, again, I don't know—honestly, I don't know how I did it because I keep asking, sometimes, people, 'Man, what is worse: really be[ing] with a community and see all your friends and lovers die, or to know that you might die next day and have nobody to tell that you might die?'[44]

## 6. Expanding and Changing Worldviews in the Face of Mortality

People living with HIV/AIDS frequently report confronting their own mortality, and often cope with loss and watching those whom they love suffer. For HIV-positive people, and those with other mortal illnesses, questions of spiritual meaning may become of upmost importance. For decades an HIV diagnosis most likely meant facing mortality—often imminent, painful, hospital-bill racking, autonomy-robbing deaths. These findings have been broken up into Section 6.1. *Answering Tough Questions* and Section 6.2. *Cultivating New Spiritual Practices.*

### 6.1. Answering Tough Questions

HIV survivors and each identity group that has been "othered" must answer tough questions that the privileged members of society have not needed to investigate. Poignantly—in spite of all the alienation, health struggles, and stigma HIV has caused people—participants often voiced feelings of gratitude, not for the illness itself, but for the ways in which it compelled them to open up in compassion, wisdom, and understanding, as a means to survive. One participant, Michael, reported:

> I think that having lived with HIV/AIDS, and having lost so many friends, and having been through all of that—I think that I have a depth of compassion that I would not have had otherwise. In that respect, maybe it's, in some ways, a gift. I mean, I would never wish it on anybody, but that is something that is—my depth of compassion, my capacity to have that kind of compassion is something that I really value and I genuinely don't think I'd have had it without those experiences.

Similarly, Robb reported that he was also able to draw upon compassion and understanding in spite of circumstances. He voiced that he indeed still carries anger and hurt from the experiences he had in his Latter-day Saints (Mormon) denomination. Robb endured conversion therapy and the cultural trappings of a patriarchal and conservative church and community. Nonetheless, he explains that his experiences motivated him to continually seek compassion, reconciliation with those who harmed him, and a deepened global understanding:

> My dad thinks I'm antichristian, and I'm just antihypocrisy and I don't like how Christianity has been appropriated by people who really don't understand what it even is. So, anyway, I'm not against it. I just—I hate how it's being used by an oppressive tribe right now to appropriate an authority and a power that doesn't belong to them.[45]

---

[43]　See Elisabeth Kübler-Ross's book *On Death and Dying* (Kübler-Ross 1969).

[44]　Jesús Guillen, interview by Kyle Desrosiers, December 20, 2019, via telephone call between San Francisco, CA and Waco, TX, transcript, Baylor University Institute for Oral History, Waco.

[45]　Robb Ivey, interview by Kyle Desrosiers, September 20, 2019, Dallas, TX, transcript, Baylor University Institute for Oral History, Waco. Interview transcript page 18.

In this way, Robb and others have been able to reclaim the spiritual framework they were raised in to find forgiveness, draw upon the values that contribute to healing and flourishing, and at the same time voice outrage at what warrants outrage.

Indeed, for Robb it was the very church structures intended—in his words—to control and force ideological conformity, that led to Robb's spiritual and personal liberation as a marginalized gay man and later as a marginalized person living with HIV. It was while on his mission in Japan that he first experienced the joy of attraction and romantic interest. He had a crush on one of his companions who was also a missionary. He connects his first crush to the love he developed for the way of life of the Japanese people:

> So, I had a crush on one of my companions. I started to see some beautiful things, and one of them was embed[ed] in my companion: his love of nature. I remember spitting out some gum sometime. It was supposed to go into a grate and it missed, and he stopped us and he says—and went, and picked it up, and put it in his pocket in some paper and said, 'We don't do that here,' and then, I look around and the country's beautiful! It's so clean and so natural, and it's because they take care of it and they love it. I was like, 'I need to learn from them.'[46]

Ron, a Black man raised Roman Catholic, experienced a similar journey out of the messages of inadequacy and worthlessness he received from the Church's doctrine—but especially the people in the Church. Ron initially responded with anger at the Church, but he was not able to find an affirming theology from within its framework, as Robb was able to do to some extent. Ron explains how he felt tremendous sadness and betrayal and confusion at the ways in which he thought that the God who was supposed to love him was said (by clergy and laity) to be punishing him for being gay and living how he did. He explains the spiritual trauma that continued to persist even after he felt that God—at least not how God was said to be in Catholicism—did not exist:

> When I graduated high school and left home ... and got diagnosed, I think I had spent so much time warding off God like, "Fuck God. There's no God. What are you talking about?" and being so angry at how I felt about myself as a result of this God that I was supposed to fear. Part of my guilt was linked to that too because I was like, "Well, maybe if I had just lived a more wholesome life, then—," and I apologized to God. I did. I remember praying and apologizing, and I even felt ridiculous at the time because I was like, "You don't owe me anything", and I had a breakdown.[47]

Yet, Ron, who is among the younger generation of men living with HIV, expresses the tremendous change in worldview he has undergone, as he has grown in compassion, empathy, and forgiveness—and continued to learn about other (and more liberating) ways to view the world, meaning, purpose, and spirituality:

> This day and age, I am a very, very different person, and it's the thing where I am very comfortable stating that I am not a religious person, and I am very comfortable talking about religion, and I respect those who have faith in a God or whoever or whatever deity they decide to meet. I think about 4 years after my diagnosis would be the time when I really started questioning like, 'Okay. I'm here. I'm clearly not going to die. What can I learn? What can I teach? How can I change me? [How] can I heal myself of the anger that I felt that affected the space around me?'[48]

---

[46] Robb Ivey, interview by Kyle Desrosiers, September 20, 2019, Dallas, TX, transcript, Baylor University Institute for Oral History, Waco. Interview transcript page 8–9.

[47] Robb Ivey, interview by Kyle Desrosiers, September 20, 2019, Dallas, TX, transcript, Baylor University Institute for Oral History, Waco. Interview transcript page 14–15.

[48] Ron Wilson, interview by Kyle Desrosiers, September 19, 2019, in Dallas, Texas, transcript, Baylor University Institute for Oral History, Waco. From interview transcript page 15–16.

Distinct from Ron, Michael decided to return to observing his childhood faith in adulthood. He explains that the impetus for the journey back to faith was the death of his devout Jewish father, with whom he had made peace only just before his death. Michael explains that though he was not an observant Jew, he decided to go to a temple in Los Angeles to say Kaddish, the prayer of bereavement, for his father:

> Because I stumbled into Kol Ami[49] that day just to say a prayer for my father—I mean, what could've been more meant to be? So, the dots all seem to connect, sometimes, in a way that I think is with all of the good, as well as with all the things that have been horrible. It all seems to connect in a way that has sort of made my life meaningful, even with the things that have been terrible, and I think that's really the full spectrum of a full life is that you have—maybe when you have the deepest of depths in despair and the heights of joy that—maybe having that full spectrum of experience is a really rich life. I tell myself that, anyway.[50]

Other participants—many of whom had lost their faith or religious practices due to experiences of stigma and bigotry—found their way home to rediscovered spirituality or found new spiritualities through different impetuses. For example, Robb explains that in addition to learning about his own and other worldviews, healthy friendships allowed him to express what he believed about meaning and purpose in the world, in order to seek compassion and forgiveness, and chart a new way forward:

> My friends were the immediate crutch I needed to reassure me, and to be there for me and not be alone. I was opening up more and more. I'm 180 [degrees] of what [where] I started. I was close-minded, from a close-minded paradigm, from people that really didn't know what spirituality was. They were just raised, 'This is what you think and that's the end of it.' You don't question because we just accept that by faith and we move on. The daily spiritual thing is that it is a continual discovery. It's amazing and it's so much more interesting than that journey, which was just kind of a dead-end, one-way street.[51]

In a world in which HIV-positive and LGBTQ people face legal inequality and social ostracization, chosen families—often friends who share similar experiences—can provide profound sources of community, in addition to creating a partnership system by which ideas, comfort, and a brand-new kind of pastoral care can be exchanged. Robb, Ron, Jesús, and Michael cite the importance of friendships in their personal development, which Jesús explains that his commitment to having friends of all kinds in all kinds of backgrounds and communities has contributed to his expansive worldview. Most participants identified the continuously unfolding and long-term nature of spiritual growth, as Robb summarizes: "Nobody knows any better than you do. It's your journey... to start where you are and continue to grow your whole life."[52]

According to Ron, and in the larger mystical tradition, the largest obstacle one often faces in their spiritual journey towards growth and peace is one's self. Ron explains the alienation he faced in America growing up poor in the Black ghetto, the alienation he felt as a queer youth in the Roman Catholic Church, and also frustration with the emphasis on beauty and achievement within the gay community, and how these factors contribute to his complicated relationship with ego. Yet, despite the emotional, physical, and institutional warfare that HIV-positive and queer people are bombarded with

---

[49] Kol Ami is a well-known LGBTQ-affirming Reform synagogue in Los Angeles, CA. It is among the first of its kind in the world.

[50] Michael Sugar, interview by Kyle Desrosiers, October 2, 2019, telephone interview between Los Angeles, CA and Waco, TX, transcript, Baylor University Institute for Oral History, Waco, TX. Interview transcript page 24.

[51] Robb Ivey, interview by Kyle Desrosiers, September 20, 2019, Dallas, TX, transcript, Baylor University Institute for Oral History, Waco. Interview transcript page 20.

[52] Robb Ivey, interview by Kyle Desrosiers, September 20, 2019, Dallas, TX, transcript, Baylor University Institute for Oral History, Waco. Interview transcript page 27.

from the outside world, Ron nevertheless asserts the significance of the internal processes needed to advance his spiritual journey:

> Once you remove that ego block … and everything comes tumbling down, it's all on the table, so then, when you start to build, the table is the foundation. So, if you're picking from what's on the table and actually learning about everything that's on the table, nothing is beneath you. Just stand on the same foundation with everything else, and you can open your mind and the experiences you can have are limitless, and an experience doesn't have to be something tangible.[53] As queer theologians have asserted[54], those standing at the margins profess that there was never anything wrong or evil about them, they who were labeled as "transgressors." Rather than being intrinsically disordered, they affirm that they are whole and good from the start. But for many, like Ron, it was a long and arduous process to begin this discovery. For Ron, it is also thanks to healthy, loving, and equitable relationships that he ate of the good fruit he needed for a better spiritual story:

> I've been with my husband now for 11 years … but for the first 3 years, we were rocky, and I was still trying to meet people, but I still had this sense of shame, and guilt and everything, so as I started to learn these things and started to try to apply some of the teachings [about Buddhist spirituality, selflessness, seeking understanding, etc.] to my life, I discovered that I started making more quality connections because I stopped looking for validation, or for attention, or for a lot of the things that we, as humans, look for.[55]

Ron reports that his husband continues to be the strongest force for healing, support, and flourishing in his life. He explains that through overcoming his ego, examining pain and trauma, practicing forgiveness and self-accountability, and seeking peace, Ron was able to produce a healthy union with his soulmate. He says that some relationships—romantic and friendship alike—needed to end to arrive at a healthier inner life. Similarly, Jesús explains that a significant catalyst for his rigorous spiritual examination was his community—when he noticed how fast so many of his friends, lovers, acquaintances, and strangers were dying and suffering. A common experience among HIV-survivors is the memory of attending a funeral once a week. Another participant remembered "losing count" of the number of friends he lost after 100. So, it is salient that Jesús, who has survived for decades, asked,

> Why am I here and these people didn't? [sic]. So, what I can tell you is I still feel... that I have to do good for others because I'm here, and I don't have to, but it's like part of me is like even if I suffer from chronic pain or whatever, I'm still here. I'm lucky to enjoy this view that I'm looking at. I'm lucky to be talking to you. I'm lucky.[56]

It is thus in summation of many of these experiences that the participants and other people living with HIV have desired to cultivate new answers to questions of meaning, purpose, and transcendence. The next section will delve into a few of the new spiritual practices and understandings that participants have discovered to in their spiritual journeys.

## 6.2. Cultivating New Spiritual Practices

In the wake of personal and collective trauma, and in most cases, a rejection by both religious institutions and their doctrinal trappings, participants sought more holistic and less fundamentalist

---

[53] Ron Wilson, interview by Kyle Desrosiers, September 19, 2019, in Dallas, Texas, transcript, Baylor University Institute for Oral History, Waco. From interview transcript page 18–20.

[54] See: (Shallenberger 1998; Guest 2006; Shore-Goss 2013; Albert et al. 2001; Tonstad 2018; Cheng 2011; Boisvert and Johnson 2012; Knauss and Mendoza-Álvarez 2019).

[55] Ron Wilson, interview by Kyle Desrosiers, September 19, 2019, in Dallas, Texas, transcript, Baylor University Institute for Oral History, Waco. From interview transcript page 19.

[56] Jesús Guillen, interview by Kyle Desrosiers, December 20, 2019, via telephone call between San Francisco, CA and Waco, TX, transcript, Baylor University Institute for Oral History, Waco. Interview transcript page 18.

answers. This search for new answers manifested itself through a variety of experiences, practices, and understanding, ranging from the mystical to the contemplative to the profoundly agnostic. For example, one participant, Robb, explained a metaphysical experience he had while still a student at Brigham Young University, which he affirms helped him grow and feel at peace in the midst of uncertainty around his sexuality and his place in the world. As he acknowledges, spiritual and metaphysical experiences are so personal, so internal, that they are hard to articulate with exactness, and sometimes, the act of articulating them results in a truncating of the intrinsic meaning and sacredness of such an experience. Nonetheless, Robb attempted to share a metaphysical, out-of-body experience he had while still in the difficult condition of being gay and different in a rigid environment:

> This is when I was in college in my dorm room and I was at my desk, and just no one was there. It was a quiet moment . . . But, just all of a sudden, I saw... this beautiful, peaceful, interstellar nursery, so quiet and beautiful . . . and immediately was this rush that felt like a hug on my insides that just said—it didn't say anything, but it was just like, 'This is you. You are this everything. This is you. It's all this. The stars just becoming what they are, and you're it, and you just feel that love. That's what it is.' That's one of my most sacred experiences, and I hate putting it into words because it was mostly the feeling of love that I got and you cannibalize it for parts when you put it into words, but it did change me. It started to open me up some more. I was so conservative from all that conditioning that when I heard—when I felt that, it exploded my mind a little bit, like, 'I don't know anything.' And out of chaos, anything's possible.[57]

For another participant, Jesús, spiritual practices and experiences certainly shaped and defined the way he understood the burden that came with being HIV-positive. However, he does not profess or cite a single spiritual experience as an impetus for new revelation and contemplation, but rather a culmination of many spiritual experiences that clarified for him over time what he refers to as a spiritual gift:

> I had always felt certain kinds of energies and certain kinds of things that happen around me, and through all my life . . . but about—I would say, maybe, about 6 years ago or something like that, many things happened. I was just arriving home. I was at the entrance in the building, and suddenly, this guy that was just passing by, he just told me, 'You know you're a shaman, right?' and I just smiled and I thanked him, but then, I went to the person who has been my acupuncturist healer and I told her what happened, and she asked me the same question; 'Didn't you know that?' I think many times, we know things, but it's very different when you grab a word and you put it in yourself and you really do accept that responsibility and that concept of that. To be very honest with you, I have many times where I feel I forget my spiritual side because I get into the earthly things, (laughs) especially when you deal with chronic pain and things like that.[58]

For Jesús, shamanism is a transgressive kind of spiritual liberation that allows him to return to the pre-Christian, precolonial spiritual practices of his ancestors, a kind of reclamation of ancient spirituality that is parallel to the reclamation of his sacredness as a queer person, in spite of the messages from religion and culture, which may suggest differently. For Jesús, he can be a different kind of shaman: deviating from binaries and expectations or norms. Since he has a queer and HIV-positive identity he has already challenged some norms and may feel liberated to challenge others:

---

[57] Robb Ivey, interview by Kyle Desrosiers, September 20, 2019, Dallas, TX, transcript, Baylor University Institute for Oral History, Waco. Interview transcript page 28.

[58] Jesús Guillen, interview by Kyle Desrosiers, December 20, 2019, via telephone call between San Francisco, CA and Waco, TX, transcript, Baylor University Institute for Oral History, Waco. Interview transcript page 1–2.

> Usually shamans are about being a healer or a warrior and still, even in this concept, you still
> have to decide who are you, and in my case, I have always felt that I'm not exactly neither
> of those two exact terms, and that's how life works. It's not just sometimes one little thing.
> For me, I always have felt that I am a bridge, a person that makes connections in many
> different ways between people and other ways of thinking or other kinds of people, and I
> love doing that maybe because I like it, maybe because it is part of me in my DNA.[59]

For Jesús, humility is a key spiritual virtue. Like Ron reported, Jesús explains that a compassion for emptying himself to serve others paradoxically empowered him and led to his own healing. Jesús lives with chronic pain due to peripheral neuropathy, and believes that spiritual practice is more important than belief, though the two are inextricable:

> I started to learn about transcendental meditation ... Every day, I still always try to at least
> take a few minutes to have my own praying, if you want to call it that way, and be thankful
> for things, and have a candle, and just relax for a moment and be one with it—whatever "it"
> is. In many other religions ... faith is a very important part of what they do, and in my case,
> I think I do the opposite. I just like to recognize that [the presence] and I turn on a candle,
> I put some music, and I just kind of be with it. When I just start writing words, or poetry
> or whatever, and I don't know where all these thinkings [sic] are coming from many times.
> I just write certain things ... and then I go to people and I tell them, "You know something?
> I just have to tell you this," and I just tell them things that I feel. I feel like if I can make
> them feel good for one second, that can create a wave of good things around, and for me,
> that's very important. I do believe a lot that anything that you do is going to create a wave,
> one way or another.[60]

In summation, it was through a combination of experiences of the heart and the head that led many of the participants toward cultivating a deeper spirituality and healthier religious orientation. They worked to synthesize a track of personal emotional healing with the desire to alleviate pain, trauma, and discord for others and the earth itself. The struggle afforded many participants with greater depths of understanding and compassion.

## 7. Coping Spiritually with Community Trauma

The HIV-positive community has been subject to collective stigma, violence, and abuse. This has ranged from social ostracization, to the apathetic, capitalistic profits-before-people response of pharmaceutical companies in offering accessible antiviral drugs, to the popular religious mythos of condemnation and apocalypse. These latter religious constructions around condemning AIDS represented a gaslighting experience of double victimization for gay and queer people. Often, the religious establishment showed little compassion for human suffering and instead blamed HIV/AIDS patients for their own situation for being "sinners" or "promiscuous." Trauma responses are divided into Section 7.1. *Coping with Collective Trauma* and Section 7.2. *Dealing with Personal Trauma*.

*7.1. Coping with Collective Trauma*

One theme common to nearly every interview was anger at the inaction of the US government and pharmaceutical companies to develop, fund, and act quickly in the face of AIDS. It was the consensus of participants that these institutions share a kind of collective culpability with collective victims—a sin that is yet to be reconciled. This inaction resulted in hundreds of thousands of premature and painful

---

[59] Jesús Guillen, interview by Kyle Desrosiers, December 20, 2019, via telephone call between San Francisco, CA and Waco, TX, transcript, Baylor University Institute for Oral History, Waco. Interview transcript page 2–3.

[60] Jesús Guillen, interview by Kyle Desrosiers, December 20, 2019, via telephone call between San Francisco, CA and Waco, TX, transcript, Baylor University Institute for Oral History, Waco.

deaths at the margins of society. Outside of the Republican Reagan administration and the capitalistic profit-first era of the 1980s which resulted in poor and inaccessible AIDS healthcare, Michael said that the most profound source of hatred toward the HIV-positive and gay community came from religious voices, particularly Christian leadership in America. In other communities, however, it is worth noting that a majority religion, such as Islam in Northern and Central Africa, represented the most consolidated source for religious rhetoric contra persons living with HIV.[61] Michael explains that HIV/AIDS only seemed like a real concern to the government once it reached "mainstream" populations:

> The White House was... at best, indifferent. And I remember once seeing something in a news broadcast that, 'Well, maybe AIDS might spread to the mainstream population. Maybe now it's time to pay attention,' and I'm like, 'Maybe? Now? Who are we? Who are we? Who are me and my friends? Why is it only now that it's spreading to, maybe, straight people that it's time to pay attention?' Also, and this is something that, I mean, I still have a lot of anger in me about it, did you ever hear of Legionnaires' disease? I don't remember where it was, but it was, like, a half a dozen or maybe 10 straight people who came down with this disease and it was everywhere on the cover of every paper and every magazine! I mean, it was the hugest news imaginable—this disease that struck about 10 people! And AIDS was now—I mean, tens of thousands of people were dying and didn't get that kind of attention, and clearly, because it wasn't happening to straight people who had little families in small towns. It was happening to people who were—who could be marginalized, who could be dismissed, who were just expendable.[62]

This incident of Legionnaires' disease revealed this observable dichotomy—the cultural conditions by which religious mythologies and moral codes were so manipulated to justify apathy and hatred of "different" groups, while the mainstream community could face much less globally consequential illnesses and be aroused toward sympathy and compassion. He—and other interviewees—noted a developing awareness that, if they did not "act up," then nobody else would act on their behalf. Later, as many historians[63] indeed have, Michael compares this complicity of the American people during the HIV/AIDS epidemic with the inaction of non-Jewish bystanders during the Shoah, which was faced by Michael's parents. For Michael, participating in corporate protests, acts of civil disobedience, and fellowship with ACT UP became a spiritual practice of protest, even before he was diagnosed as HIV-positive:

> That's what was the source of my activism, at least, was I decided that it was a matter of life and death to put AIDS in the faces of people who didn't want to know about it, to make noise, and to never shut up. And I remember when I started showing up for demonstrations and things like that, that many of the people that were there at those demonstrations—I mean, I could see on their faces that this was—it was too late for them. So many of those people were so sick that it was obvious that it was too late for them. They were doing this for somebody else. So, I don't know how to describe the combination of anger, outrage, upset, sadness, and inspiration. I can never forgive the religious leaders and the government leaders who twiddled their thumbs and did nothing while my entire community died.[64]

Due to the vitriol, hatred, and ignorance of those in power—who happened to generally be white Protestant Christians—Michael, and many others interviewed, have understandable anger at the Christian faith and the rhetoric of harm many of its leaders spread regarding HIV and LGBTQ people.

---

[61] (Balogun 2010; Haddad 2011).

[62] Michael Sugar, interview by Kyle Desrosiers, October 2, 2019, telephone interview between Los Angeles, CA and Waco, TX, transcript, Baylor University Institute for Oral History, Waco, TX. Interview transcript page 20–22.

[63] See: *Reports from the Holocaust* (Kramer 1997).

[64] Michael Sugar, interview by Kyle Desrosiers, October 2, 2019, telephone interview between Los Angeles, CA and Waco, TX, transcript, Baylor University Institute for Oral History, Waco, TX. Interview transcript page 21–22.

Christian people, the Christian religion[65], and even talk of Jesus have been linked to apathetic medical responses and discrimination in the mind of many survivors, and so they are the cause of trauma, pain, and distress:

> Most of the hate that I have had pointed at me, religious-based hate, has come from people who call themselves Christian... this kind of hate in America, anyway, expressed in the name of Jesus, and that, to me, is sickening. And I have to say that to this day, there's a lot of things that haven't changed in that respect, but a lot of that fueled—and the coining of the expression 'family values', which, of course, meant—that was code language for hating people who didn't live the way that they liked, the way they did; people who were gay or lesbian. 'Family values' was language that expressed hate for LGBT folks and this whole idea of AIDS being some kind of divine retribution punishment from God, and even at the time that then-president Ronald Reagan—he didn't mention the word 'AIDS' for years, and when he did, it was in the context of, 'This is something that people could avoid by living a moral lifestyle,' so I not only—I mean, I had a lot of reasons for not, at that time, having any religious or spiritual activity in my own life because I only witnessed so much hate, so many expressions of hate coming from religions communities.[66]

Notably, all three Jewish participants interviewed for this archive as of April 2020, (some of whom are not included in this abridged article), had all maintained, developed, or rediscovered some kind of Jewish identity. On the other hand, out of the seven participants who were raised Christian, only the two were still Christian-identifying (Metropolitan Community Church and United Methodist; one had also converted to Judaism)[67]. Several reported that these religious changes were due to the extensive stigma and bigotry inordinately associated with Christian denominations in America against HIV-positive people and homosexuality.

About the relationship between religious stigma and HIV/AIDS, a Waco, Texas pastor, the Reverend Charley[68], who is not HIV-positive, but offers pastoral care to many HIV-positive survivors at his Metropolitan Community Church (MCC) parish through support groups and a food pantry, voiced profound hurt and disgust at the way the Christian gospel has in some instances been used to justify negligence, violence, and apathy by those in power. Due to this, Charley offers a compassionate Christian response to those who can no longer trust Christianity, or any religious tradition because of the harm it caused them and those they love(d):

> All we knew was our friends were dying, and so, there was a lot of questioning, and certainly, when you're harangued by religious fundamentalists who are saying, 'It's because God is killing you and for good reason, 'yes, there will be people who abandon religion, and I can understand why that would be the case. That's still the case today, not necessarily because of HIV, but there are still religious fundamentalists who oppress the LGBT community

---

[65] In *The Band Played On* (1987), Schilts is explicit in his criticism of the Evangelical Church(es), Jerry Falwell's empire, and the "Moral Majority" movement for their apathy toward the human suffering of the AID epidemic. A recent podcast from America Media, *Plague: Untold Stories of AIDS & the Catholic Church* (O'Laughlin 2020) tells the history of the relationship between the American Roman Catholic Church and HIV/ LGBTQ activism. Lastly, others, such as Mark Thompson in *Gay Spirit: Myth and Meaning* (Thompson 1987), have also noted a link between the harm organized religion constituted for HIV survivors and queer people, but also a newfound spiritual fervency and purpose discovered in queer activism, such as ACT UP.

[66] Michael Sugar, interview by Kyle Desrosiers, October 2, 2019, telephone interview between Los Angeles, CA and Waco, TX, transcript, Baylor University Institute for Oral History, Waco, TX. Interview transcript page 22.

[67] Please see interview transcript corpus at Baylor Institute for Oral History database for information about participants' self-identification.

[68] Charley Garrison, interview by Kyle Desrosiers, October 29, 2019, in Waco, Texas, transcript, Baylor University Institute for Oral History, Waco.

and because of that oppression, they walk away from any form of organized—especially Christian—religion, and all I can say is, 'I understand.'[69]

Other interviewees, such as Robb, who came of age in the early 1980s, explain that they never knew a time in which they could enjoy dating, sex, and relationships without a climate of fear, death, and suffering. Robb, who grew up sheltered and Mormon, said he never knew a time without HIV as a looming threat to the gay community:

> So, I was about [21 years old] when I came out and—let's see. That was early eighties, so AIDS was just starting to hit the news and people were scared, and I grew up—I did not know a time when we weren't terrified of AIDS, and so, I immediately knew that it was dangerous, sex was dangerous, and that you had to protect yourself, and it was not to be messed with or trifled with. You were very careful [during sexual intercourse] and respectful of yourself and others.[70]

Yet, in this impossible climate of fear, love persisted. Gays and lesbians joined ACT UP and advocated, held weddings—in the eyes of God and not the state—and attended funeral after funeral to honor the deceased among them. Romance persisted, soul mates and lovers nonetheless found each other, and built love and lives. However, as many interviewees explained, gay relationships came with many challenges heterosexuals never had to confront. Robb remembers falling in love with his first partner and eventually moving in together. Because of the inherited shame and fear, his partner felt he needed to keep his HIV-status a secret:

> He didn't tell me he had AIDS, but he was also very protective about what we did sexually... I remember when he finally told me when he couldn't hide it anymore, how he just collapsed on the floor just riddled with shame, and guilt, and the horror of it all, and the fear that I would leave... We went through a marriage ceremony between friends, but, of course, it wasn't legal. It was just something we did that we wanted to do with each other. And then, he died.[71]

Another participant, Jesús, who also comes from the older generation of survivors included in this project, remembers coming to the United States from Mexico, with little experience of any connection to the gay community, only to see it soon decimated. He remembers, "It was almost part of my welcoming committee to this country." Jesús remarked on his confusion and periodic guilt in having survived the epidemic while so many others did not:

> 'How is it that you survived, and all these people didn't?' We didn't, period. So, every day until 1996 or 1998, almost until 2000 when they could tell if medicines was better and everything, the reality is that we were still thinking, I might die tomorrow. People were getting sick of pneumonia, or some cancers, or this and that and die in a few days or weeks, sometimes, and I didn't took [sic] medications for 15 years.[72]

### 7.2. Dealing with Personal Trauma

The HIV-positive community has been and continues to be subject to societal stigma and institutional barriers to adequate social services. The complexity of this multifaceted trauma continues

[69]  Charley Garrison, interview by Kyle Desrosiers, October 29, 2019, in Waco, Texas, transcript, Baylor University Institute for Oral History, Waco.

[70]  Robb Ivey, interview by Kyle Desrosiers, September 20, 2019, Dallas, TX, transcript, Baylor University Institute for Oral History, Waco. Interview transcript page 14.

[71]  Robb Ivey, interview by Kyle Desrosiers, September 20, 2019, Dallas, TX, transcript, Baylor University Institute for Oral History, Waco

[72]  Jesús Guillen, interview by Kyle Desrosiers, December 20, 2019, via telephone call between San Francisco, CA and Waco, TX, transcript, Baylor University Institute for Oral History, Waco. Interview transcript page 15.

to be largely misunderstood by mainstream society. The previous section addressed societal-level, collective experience among many of the participants—whose identities represented some overlaps and distinctions in terms of their reported experiences. In addition to larger-scale, more universal experiences, such as anger at governmental inaction, many participants also reported instances of personal trauma, whether in relation to their romantic partners, family members, or others in the HIV-positive community and gay community, who, in coping with their own horrors and trauma, contributed to the suffering of others. For some, there is a phenomenon wherein the crucible of tragedy instills in others anger and self-centeredness rather than an opening up of the compassionate mind.

One participant, Robb, shared a memory about how he became HIV-positive. He was unknowingly infected by someone else who intentionally spread the virus to him. The man who he later came to realize spread HIV to him, acted with incredible cruelty. So, it was just months after his partner died of HIV that Robb had this experience. He began to meet new sexual partners:

> With the very first person I was with, I was very careful. We would get to an exciting point [in sexual intercourse] and he would act like he was going to do something, and I'd say, 'No. Put a condom on,' and then, we'd go back to kissing and getting excited, and I'd say, 'Oh, you got to put the condom on,' and then he acted like he put a condom on, and I thought everything was fine, but at the end of that session, he said, 'I took you to the dark side.' And so, I did not understand that. I just—I was in the afterglow—didn't even think about it until a couple weeks later when I went for testing, and then, they told me, 'Oh, you're positive now.'[73]

Robb responded to this act of cruelty with compassion and forgiveness, which came gradually as he began to internalize his new reality. He struggled to understand why another human could be so tormented that he could inflict a similar fate on others:

> I have no idea. I would like to know why. He was a beautiful, interesting man. I don't know what inside him made him do that. I worry about how many other people whose lives he disturbed that way. I have no idea. I looked for him immediately—you know, 'Why did you do this?'—and I could not find him. Passing through town on your mission of death? What the hell? (laughs) I thought I was going to die. I had just buried my partner.[74]

For another participant, Ron, one of the greatest traumas linked to sexuality came from his first sexual experience—an incident in which he was raped as a child:

> My first experience with a man was by force and I was nine, and so, that was a traumatic experience, and especially now, after living so much life, I realize that traumatic experiences are probably some of the things that shape us more so than anything else in our lives. I just sort of hid behind being a jock and being smart, and I figured as long as I presented that and not the other part, it's going to be fine.[75]

Ron also voiced that his relationship with his former partner—one that became tremendously unhealthy and emotional abusive—was based on dishonesty. This directly relates to both HIV-positive and LGBTQ experiences; research suggests of partner abuse and relational immaturity is symptomatic of societal marginalization, which have resulted in adults with poorer mental health outcomes than their

---

[73] Robb Ivey, interview by Kyle Desrosiers, September 20, 2019, Dallas, TX, transcript, Baylor University Institute for Oral History, Waco. Interview transcript page 15.

[74] Robb Ivey, interview by Kyle Desrosiers, September 20, 2019, Dallas, TX, transcript, Baylor University Institute for Oral History, Waco. Interview transcript page 15.

[75] Ron Wilson, interview by Kyle Desrosiers, September 19, 2019, in Dallas, Texas, transcript, Baylor University Institute for Oral History, Waco. From interview transcript page 7.

more privileged heterosexual counterparts.[76] Queer identities have been stigmatized for generations, and people in LGBTQ relationships have not been given the same kinds of healthy encouragement and positive representation that heterosexual people are socialized to understand. The product of the collective trauma for being marginalized for queer identities and HIV-status may drive some to destructive responses by which individuals choose to spread their suffering to others. Ron, like Robb, shared an episode from this life in the months immediately following his diagnosis, as he experienced this particular struggle:

> I felt the responsibility to tell this person that I was dating, 'Hey, I got a positive diagnosis,' but I had to do it quickly. I was like, 'What if I give it to him?' So, I told him, and he sort of cut me off and I just thought of myself as shit ... He died 2 years later, and the friends that introduced us told me that he was positive and he was never on medication, and they were very upset with him, but didn't feel like it was their place to share that with me while he was alive. But, he didn't take care of himself when he spiraled and didn't take any medications, and they think that he was in a dark enough place to want to die and want to not do what he had done, apparently, to not just me, but other people. Here I was walking around and I was already processing my own anger issues, and here I am now, just learning that I didn't affect someone else's life, their life affected mine, and this asshole cut me off because he knew what he had done and he let me walk around thinking that I had done something to someone, and it was that same sense of guilt that you get from religion where you walk around thinking you've done something wrong to someone just by living.[77]

Jesús, who was a Mexican national living in California in the 1990s, expressed the unique fears and challenges he had because of HIV and his American residency status:

> When I became HIV-positive, mijo, I didn't have papers here. So, there was the amnesty happening in this country and they were doing the blood test to see if it will affect you for the amnesty, so I knew already I was HIV-positive and I will be thrown out of this country, so that was the first time ever that I had to come out to someone about me being HIV, and once again, I'm very thankful to this friend—he did his blood test for me because in those days HIV-positive people were not allowed to immigrate to this country.[78]

In America, Jesús found support and hope in his Latino gay friends. He joined a gay men's chorus and was one of the founders of a group called ÁGUILAS, a community group for gay/queer Latinos. He says, that despite the lack of visibility or positive representations of gay people in his Mexican culture—and also in the US culture in those days—he misses one key value from his culture:

> In other countries, the concept of family and group is higher than here. Here, it's about the individual, and it's a great thing. I learned a lot from that, coming from a society that family, sometimes, is too much, but it should not be one or the other. It should be a combination of both. And right now, again, this community of people growing older with HIV and survivors—they're feeling left out.[79]

---

[76] Psychologist Alan Downs has devoted much of his career meeting the needs of patients who were gay men coping with the trauma instilled in them by society, institutions, and family. One phenomenon he reports is that whereby some respond to tragedy and marginalization by becoming meaner and doing more harm to others–instead of having more compassion. See: *The Velvet Rage* (Downs 2012).

[77] Ron Wilson, interview by Kyle Desrosiers, September 19, 2019, in Dallas, Texas, transcript, Baylor University Institute for Oral History, Waco. From interview transcript page 12.

[78] Jesús Guillen, interview by Kyle Desrosiers, December 20, 2019, via telephone call between San Francisco, CA and Waco, TX, transcript, Baylor University Institute for Oral History, Waco. Interview transcript page 27.

[79] Jesús Guillen, interview by Kyle Desrosiers, December 20, 2019, via telephone call between San Francisco, CA and Waco, TX, transcript, Baylor University Institute for Oral History, Waco. Interview transcript page 20.

The shadow side to community-oriented culture (i.e., as is often experienced in Latin America) is that gays, women, trans people, and racial/ethnic and other outsiders can easily be alienated and neglected. On the other hand, the shadow side to individualistic Western culture, is that there is a great emphasis on personal success and well-being, sometimes at the expense of others who are less "useful" or desired as friends, partners, and colleagues. Both have drawbacks and both affect minority groups in distinct ways. This latter problem has been a challenged in the gay community, which has historically been obsessed with youth[80]—but is also a problem in larger mainstream Western culture.

Additionally, another participant explained the nuances of his Black identity in relation to being HIV-positive and gay—especially coming from a low-income Black community. He believes his community prioritized rigid religion because they had so little else that could be taken for granted in a society where they had to work so hard to make ends meet, and deal with the daily uncertainty of negligence from the government, bad education systems, and a lack of institutions that their white and more affluent counterparts possess.[81] Ron explains that his Black identity both alienated him from what he saw as heterosexist culture in his Black community, and also the white dominant culture. Due to these conditions, faith and religion became very complex in Ron's struggle:

> Especially because being Black and being from the ghetto ... where you're poor and the only thing you have to hold onto is faith, and religion, and everything—they stick to religion because that's the one thing that they do know, and so, when you don't follow what they see as the word, or the book, or his way or anything like that, you're looked down upon: 'Why am I in such fear of something that is supposed to be love, and good, and guide you?'[82]

Ron sees a link between the experience of Blackness in America and marginalization for being LGBTQ and HIV-positive. It is this marginality that first taught him how to cope with a status quo stacked against him and those like him:

> So, coming from a marginalized community before I even knew what gay was, or what it meant, or that I was gay, I can say that you question everything. You question everything because you don't trust.[83]

## 8. The Spiritual Project of Dealing with Loss and Death

Most participants in this project, and those in the world who live with chronic or life-threatening illness or among loved ones with these conditions, have to deal with loss and mortality. Due to the experiences of losing partners and friends, especially for HIV-positive and gay people in the 1980s and 1990s, as well as the necessity of confronting their own mortality, most of the participants reported undergoing rigorous evaluation of their worldviews, spiritualities, and cosmologies, particularly with respect to how they understood, processed, and coped with death and suffering.

One example of a collective spiritual mourning endeavor was the AIDS Quilt. The AIDS Quilt was an American project of pluralist, democratic mourning that brought the spiritual and memorial components of the HIV/AIDS crisis to the forefront. It served to raise awareness, understanding, and support within the community outside the HIV/AIDS community and the gay community.[84] The AIDS Quilt embodied a kind of strength that arose in the wake of loss and the ambivalence of American society and government. It was a communal endeavor, both a religiously pluralistic and uniquely democratic method of mourning, activism, and strivings for reconciliation—all rolled into one in the American secular sphere.

---

80    See: *Straightjacket: How to be Gay and Happy* (Todd 2012).
81    See: *The Tragic Vision of African American Religion* (Johnson 2010).
82    Ron Wilson, interview by Kyle Desrosiers, September 19, 2019, in Dallas, Texas, transcript, Baylor University Institute for Oral History, Waco. From interview transcript page 6.
83    Ron Wilson, interview by Kyle Desrosiers, September 19, 2019, in Dallas, Texas, transcript, Baylor University Institute for Oral History, Waco. From interview transcript page 20.
84    (Cleworth 2012).

In the course of one interview with the participant Michael, the AIDS Quilt was a prominent element of his spiritual journey. He explains what the AIDS Quilt was, and what it meant at the time of its execution:

> The AIDS Quilt was started in San Francisco. The quilt is like—people create quilt panels three feet by six feet, which is about the size of a grave, with somebody's name on it to commemorate the people that have died of AIDS and we wrote their name. Some people sew into these quilt panels personal effects and mementos or things that represent who that person was, and these panels are all stitched together, and so, they could be laid out by the tens of thousands. The last time it was displayed in Washington, it stretched from the Washington Monument to the Lincoln Memorial. It's almost incomprehensible how huge this thing was.[85]

The size of the AIDS Quilt, and number of its panels serve as a visual reminder of the sheer volume of humans who suffered and died due to AIDS. According to Michael, it has not been displayed in its entirety since the 1990s, as it has now gotten too big to fit in one location, even the National Mall. Today, organizations, such as schools, order "twelve by twelves" of panels to be displayed for occasions such as World AIDS Day. The AIDS Quilt, contemplative, quiet, and with all laid bare, stands in contrast to other demonstrations, such as the ACTUP protests inside and outside of churches such as St. Patrick's in New York. Both serve different and necessary purposes. The AIDS Quilt had a strength that even those with hardened hearts could not ignore: these panels do not only represent somebody who died, but somebody who was loved. It was mostly in memorial of gay American men taken by the virus. It was at this Quilt that Michael experienced a profound moment of reconciliation with his father, as they both knew they would each be confronting mortality—Michael had HIV and his father was aging:

> And so, when that last display of the quilt occurred in Washington in 1996, my dad drove from Pennsylvania to meet me at the quilt. So many of my friends are represented on the quilt. I mean, I lost, during those years, everybody from the closest friends, acquaintances; I lost track. I lost over 100 friends. When I went to the quilt as a volunteer, my dad drove from Pennsylvania to Washington to meet me there, which was about 100 miles, and that was—to be there to meet me—I told him how important it was for me, but that he drove there to the quilt to meet me was such a beautiful expression of support and love.[86]

Michael explained that he was "consumed with grief [and] so deeply depressed that I couldn't even imagine that I'd ever be able to claw myself out of it."[87] He said that the words his Holocaust survivor father spoke to him at the AIDS Quilt changed his life: "Michael, you are never going to get over this, but you must learn to live with it so that you can go on with your life."[88]

In the months and years that followed, Michael explained that his relationship with his father, who was a "very spiritual and religious man" grew stronger for the first time in years. This played a significant way in the role Michael made sense of the carnage around him due to AIDS and coped with collective trauma. He shared a letter from his father in 1996, from just before his death:

> "My dear son, I want you to know that I'm thinking of you every minute of the day and night and I'm praying that in the near future, there will be a total cure for this unfortunate

---

[85]  Michael Sugar, interview by Kyle Desrosiers, October 2, 2019, telephone interview between Los Angeles, CA and Waco, TX, transcript, Baylor University Institute for Oral History, Waco, TX. Interview transcript page 12.

[86]  Michael Sugar, interview by Kyle Desrosiers, October 2, 2019, telephone interview between Los Angeles, CA and Waco, TX, transcript, Baylor University Institute for Oral History, Waco, TX. Interview transcript page 13.

[87]  Michael Sugar, interview by Kyle Desrosiers, October 2, 2019, telephone interview between Los Angeles, CA and Waco, TX, transcript, Baylor University Institute for Oral History, Waco, TX.

[88]  Michael Sugar, interview by Kyle Desrosiers, October 2, 2019, telephone interview between Los Angeles, CA and Waco, TX, transcript, Baylor University Institute for Oral History, Waco, TX

sickness that is snuffing out the life of so many young and talented people in our country and the entire world. Michael, please be strong and make every effort to overcome all the odds, and I always pray that you will reach an old age, and when I'm not here anymore, you'll tell everybody, 'My father told me that.'"[89]

Ultimately, it was through the moving experiences Michael had at the AIDS Quilt, and later in reconciling with his father, that he was able to begin figuring out how to "live again" after the epidemic. Others sought to find a way to learn and a new purpose in light of the carnage and suffering, and the cruelty of American society and politicians of the time. Some found meaning and purpose through service, vocation, activism, or other channels. Michael also explained that—in spite of his fear of Christians engendered by their mistreatment toward him—he saw some displays of love, mourning, and solidarity in Los Angeles in the midst of those days:

> A local church in Hollywood, the Hollywood United Methodist Church—to this day, they have a big, big red ribbon on their building that's been there for years, but when the AIDS quilt was established in the eighties, that church set up quilting workshops to help people make panels for the quilt and, more than that, to help people cope with just irreconcilable grief. I mean, what they did for many people—I'm getting choked up even thinking about it. What they did for so many people was so loving, and they were attacked by religious communities. They were attacked for reaching out to people who were grieving because of the losses from AIDS. They were attacked viciously because it was a church that reached out to people that didn't fit in.[90]

The Reverend Charley is a pastor in the Metropolitan Community Church, and was so distressed by the AIDS crisis in his community that he found a calling in building a church that served the neglected, marginalized, and infirm. Charley renounced the apathetic conditions prevalent in American institutions of the day, and instead professed hope in a better story. Charley's theology is of deep Christological significance, resonating with those who imagine Christ as the icon of all suffering and pain, the victim for all victims:

> In general, the response was either dead silence or it was thought to be God's punishment to the gay community because at that time, most of the people that we know who were dying or who were infected were gay men, and also, the prostitutes and drug users were—it was showing up there too. The MCC church had been around since 1968, and so, what, maybe about 15 years or so before the actual—before HIV started showing up in the country, and I remember thinking that we believe within our guts that HIV was not a death sentence from God, but still, there was that thing of, 'But what if? Why is this only happening to us?' it seemed like. And so, HIV actually took a great toll on the MCC denomination, both within members of the church, as well as pastors. There are a great number of people who died in the early days of the epidemic. We became known as 'The Church with AIDS'. We were the church with AIDS, and we embraced that identity, believing that we are the body of Christ and that Christ had AIDS at that time.[91]

To Charley, simplistic understandings of theology and scriptures were not enough to cope with the level of suffering and death the LGBTQ community was facing. To him, the Christ mystery was

---

89  From the words Michael shared in the audio interview, as transcribed by the Baylor Institute for Oral History. Quotes are used here to indicate that Michael spoke (read) these words during the interview. May be accessed at the Baylor Institute for Oral History. Michael Sugar interview transcript page 14.

90  Michael Sugar, interview by Kyle Desrosiers, October 2, 2019, telephone interview between Los Angeles, CA and Waco, TX, transcript, Baylor University Institute for Oral History, Waco, TX. Interview transcript page 11.

91  Charley Garrison, interview by Kyle Desrosiers, October 29, 2019, in Waco, Texas, transcript, Baylor University Institute for Oral History, Waco.

everything but simple, and Jesus Christ was subversive to the earthly status quo—just like Charley, a gay, HIV-negative minister who represented a "church with AIDS." After spending time in his hometown of Baton Rouge as an ACT UP activist and working closely with gay and lesbian pastors in the MCC church, Charley explained that a mentor of his, a lesbian minister-activist, first pointed out his call to ministry. Charley was working with AIDS patients in a hospice, providing pastoral care already, so it only made sense to get formally ordained. Charley explained that his care for the sick and dying taught him more about faith, meaning, and purpose than theological tomes ever had. It was through pastoral care and service that he created a new meaning, in spite of the great tragedy and horror of losing hundreds of friends to the epidemic. Instead of fear, he chose compassion:

> I remember . . . whenever I would go into the hospital, for instance, to visit somebody, at that time, you couldn't just walk into the room. You had to gown up and you had to put masks and everything on, and that had to take place in a changing room before you actually went into the room, so you would go into that changing room, you'd put on the gown, you'd put on the masks and everything, and then, you would go into the patient's room. The first time that I ever did that, I realized that it was nonsense. I was not going to do that to the patient. I just threw it off, took off the gown and everything in the room. I also did full-body massages for a lot of the HIV patients; people who had lost touch—literally lost touch—with other human beings. Touching someone like that was considered a risk. I say all of this to answer your question. How did that affect my faith? I basically told God, 'If you're going to kill me, take me down, because I'm not doing wrong by doing this.'[92]

As some social conditions and medical advances have changed things and other status quos continue to persist, Charley has spent the last decades since focused on HIV ministry. To him, there is nothing more sacred than caring for those on the margins:

> It instilled with me a stronger faith, I think, in a God who cared, and loved, and was on the side of the marginalized, and wanted us to do the same thing, so it really did inform my faith a lot. Also, spending time with people who were dying—I liked it. I was—I felt it to be a powerful experience, enough so that even now, I'm a hospice volunteer. You can't spend time with somebody who's facing their last days without also dealing with your faith in that, and more times than not, my faith was made stronger as I watched these people enter their last days, so it did have a strong impact on my faith.[93]

Michael explains that a similar role was performed by his spiritual advisor, the well-known spiritual leader and lesbian activist Rabbi Denise Eger of Los Angeles, and what her new take on faith community meant to him:

> I mean, after having experienced so many expressions of hatred coming from communities of—quote, unquote—faith, to walk into a place that was so loving and welcoming, where there were so many other gay people—and when I say gay, I mean that all-inclusively LGBTQ, all the alphabet soup—and the rabbi, Denise Eger spoke so eloquently about things that were relevant for me, and what she did and still does, by the way—I still go to Kol Ami. I've been a member now there for over 20 years—what she still does is she makes everything relevant. All of her sermons, all of her—when we talk, when we pray or we read from the Torah, she explains what these things mean in a way that is relevant to my life, in a way that is relevant to our times, and in a way that sort of, I think, inspires me to consider, 'What is

92    Charley Garrison, interview by Kyle Desrosiers, October 29, 2019, in Waco, Texas, transcript, Baylor University Institute for Oral History, Waco. Interview transcript page 8.
93    Charley Garrison, interview by Kyle Desrosiers, October 29, 2019, in Waco, Texas, transcript, Baylor University Institute for Oral History, Waco. Interview transcript page 8.

my purpose? What am I called to do? What is my role in the world?' which also is, of course, providing answers to all of those questions about, 'Why am I still here?'[94]

Rabbi Eger hosts a monthly Shabbat lunch with HIV-positive community members who are Jewish and non-Jewish alike, as a kind of support group. Michael identifies this as the first faith-based HIV support group in the country. Michael says that Rabbi Eger embodies what a faith leader should, because she is in the world, living, serving, and struggling alongside people. For this, he trusts her, and is able to pick up again the pieces of his broken faith:

> She was an activist in the HIV community like me since the beginning, so she helps that I'm a member of a synagogue where there's a rabbi who really understands my experience because she shares a lot of it. She did hospital visits, and she did funerals for many, many, many people during the worst of those AIDS years, so I know that when I—I mean, intuitively, that she gets it. She understands it completely. I feel supported in that by my synagogue that sort of embraces those kind of values of social awareness, activism, really using your voice, being called to make the world a more just place ... That is at the core of my Jewishness.[95]

Ron, a participant who does not identify as religious, also finds meaning and purpose in activism and the service of others. Activism and service were recurring activities by which participants coped with loss and tragedy. For Ron, the basis of his ethical code is a focus on making the present and future better, not dwelling on the loss and suffering of a past that cannot be changed. Ron sits on the board of an organization called Team Friendly DFW, an organization in the Dallas-Ft. Worth area with the mission of "kill[ing] the stigma by educating people regarding how to talk about HIV, to actually help people process the information that they probably already possess—or, at least, expand on the information that they possess.

According to Ron, the group is sex-positive, all-inclusive, and nondiscriminatory, not a "gay men only group." He explains that this group is antithetical to the way he saw organizations and institutions discuss health and sexuality when he was growing up. Shame and stigma do not work to save lives, he said, but instead, an abundance of information and nonjudgmental care does. He says he was once more self-centered, and also depressed, before contracting HIV, and eventually forming the healthy relationships and eventual marriage that he now has. He explained that he discovered a vocation for building supportive communities, because of the significant role his support system played in his life:

> The things that were most important to me at the time were the things that made me feel a sense of family or belonging, and that was performing. I was a dancer. I performed, I went on tour with the drum corps, and I taught and performed in winter guard, and those were things that were seriously important to me, and so, my experiences of being able to go to different cities and meet different people from different regions of the world—those things were really important to me because those people accepted me for me because we had something in common, and there were no expectations placed on me other than to take care of my business in rehearsals and on the floor, and then we can all go and be the same tragic messes that we wanted to be together at the end of the day, so that was great for me.[96]

Jesús, who lives with chronic pain in addition to HIV, explains that he also finds meaning working for and alongside others for the common good:

[94] Michael Sugar, interview by Kyle Desrosiers, October 2, 2019, telephone interview between Los Angeles, CA and Waco, TX, transcript, Baylor University Institute for Oral History, Waco, TX. Interview transcript page 16–17.

[95] Michael Sugar, interview by Kyle Desrosiers, October 2, 2019, telephone interview between Los Angeles, CA and Waco, TX, transcript, Baylor University Institute for Oral History, Waco, TX. Interview transcript page 16–18.

[96] Ron Wilson, interview by Kyle Desrosiers, September 19, 2019, in Dallas, Texas, transcript, Baylor University Institute for Oral History, Waco. From interview transcript page 8–9.

I have tried socially and everything to really focus in the areas of other people, HIV long-term survivors, and people growing older with HIV. By now, I am the founder of the HIV long-term survivors' group in social media. It's the largest group around the world. I just came back from the International Pain Summit in Los Angeles. I work in a local level, national level, and international level to try to keep motivating, and giving information, pass information, and to—at least that people can acknowledge the presence of this community of survivors that—and I'm one of them, but I'm considered, once again, one of the lucky ones that somehow is still able to travel, to go around, to work around. There's so many people isolated and lonely in their homes that have kind of given up. There have been more suicides, and even through social media, we've been able to stop a few.[97]

Jesús explains he found purpose in activism and in his independent spirituality, though not in a traditional faith community. He finds a calling, as a marginalized person himself, in "speaking for so many people that have no chance to speak up because they're very shy or very sick already to really speak up. Lastly, he explains that he connects this very practical service to the transcendental reality he has discovered for himself. He views his relationships with others as a flow and exchange of energy, whether it is positive or negative. Jesús has been able to find healing and rebuild a new sense of meaning through focusing on generating positive energy the best way he can:

It's almost like if I just feel like I can send it, I can. I really can share it, and especially if I can feel—if I had those people sick around me or things like that, so that has been very important even in the International Pain Summit that everybody in the building was people with pain. It's like even if I can make someone smile—or I'm very touchy-feely with a hug or even a just brief touch—like one of the ladies—she was like, 'Oh, I'm taking all your energy,' and I said, 'No, I have enough. Don't worry,' because she was hugging me tightly.[98]

## 9. Hopes and Anxieties about the Future: From Spiritual Awakening to Ethical Healthcare

The final theme surveyed was participants' perspectives on the future of the collective response to HIV/AIDS in of America and the world. Participants gave varied responses that ranged from concerns about HIV care for the most marginalized groups—e.g., sex workers, ethnic minorities and those abroad, the homeless, and drug users—to worries about how citizens navigate political conversations and the future of the environment.

Robb, almost 60, was one of the older participants, and remembers the state of the gay community before, during, and after the HIV/AIDS epidemic. Despite many positive changes, he professed a mournful attitude to certain changes in the way HIV once brought a community together:

Our culture lost something really beautiful. (becomes emotional) We really lost a sense of community, because that pain really knit us together and there was—lesbians came and we were so together helping each other, and we really had a strong community based on caring. We had so much wisdom. We had the older gays that knew things and were passing it on helping each other, and we lost all that. I feel sorry that you didn't get to have that, and that's probably why I'm doing this because I want to share a little bit that you may not get anywhere else.[99]

---

[97] Jesús Guillen, interview by Kyle Desrosiers, December 20, 2019, via telephone call between San Francisco, CA and Waco, TX, transcript, Baylor University Institute for Oral History, Waco. Interview transcript page 18–19.

[98] Jesús Guillen, interview by Kyle Desrosiers, December 20, 2019, via telephone call between San Francisco, CA and Waco, TX, transcript, Baylor University Institute for Oral History, Waco. Interview transcript page 24.

[99] Robb Ivey, interview by Kyle Desrosiers, September 20, 2019, Dallas, TX, transcript, Baylor University Institute for Oral History, Waco. Interview transcript page 21–22.

In the loneliness and stigma of the HIV/AIDS crisis, in the face of death and illness, Robb saw good. It is a very spiritual endeavor to perform this kind of introspection and find what is good within the dark times, and even mourn the absence of what once was:

> It made us very strong because we had nothing to lose, and we also knew if we didn't fight, that it was not going to happen. We had to do it. It made us strong and powerful. It was a terrifying and thrilling time to be in the gay community.[100]

Jesús, also from the generation of older survivors, was concerned by the lack of intergenerational LGBTQ solidarity, which he views as partially responsible for risky behaviors seen among young people today, misinformation, and the persistence of HIV/AIDS as an American health concern. He connects the values he wishes the gay community maintained with wisdom he hopes to apply to the rest of his life as a human being:

> I feel that we're failing in [doing] the whole intergenerational thing. Youngsters nowadays don't even know the history of what happened or how it happened. They just think that you can just take a pill and nothing is going to happen. The reality … we are the only proof of what might happen to you in 20 years. Everything that someone might tell you now—'No, nothing is going to happen,'—ask them to show you what is the proof of that. We should communicate more between generations, because otherwise—but it is part of that also, in this country, the ageism is pretty high in comparison with other countries. Here, as soon as you start losing your youth and beauty, then that's like—it's like 'scene', almost.[101]

Jesús expresses that experiences with death, illness, and bigotry have taught him the significance of recognizing the intrinsic value and humanity of every person—especially those who face other forms of marginalization. Similar to how Michael expressed the connection between the Holocaust of European Jews and the HIV/AIDS crisis constituting a genocide for gay men, Jesús agreed that there are crises today which are forgotten and neglected. He believes this is due to of a lack of values, societal apathy, or a willful neglect by those in power:

> Mijo, I'm not falling apart, but yes, I am an elder and I still have things to give to society. And also, on the other side, once again, don't call the older people ignorant, or stupid, or that they don't know what they're doing or all these things. Come on! Really? There's all kinds of people. Give a chance to every person. Now, with what's going on politically and everything, things are getting worse for many people in many ways and many things in other countries are very bad to get medicines and other things. And just the last thing, mijo, is that we are doing, right now, with [the] opioids epidemic … becoming, once again, another area of stigma and we have to be very careful with those things because the HIV people—they know how it hurts when someone talks badly about you, or don't want to touch you or don't want to be with you, so whatever disease it is, just let's be careful. Don't go into judgment without knowing or with not separating that if somebody was a whore or if somebody was just a one-time deal—don't judge too easily.[102]

Jesús was among other participants who identified similar experiences of apathy and moralizing among the current opioid crisis in the American postindustrial Rust Belt. Though interviews were conducted before the 2020 Coronavirus, there are direct similarities to this pandemic, including the ways in which health and identity, access to healthcare, and respect for scientists are politicized.

---

[100]　Robb Ivey, interview by Kyle Desrosiers, September 20, 2019, Dallas, TX, transcript, Baylor University Institute for Oral History, Waco.

[101]　Jesús Guillen, interview by Kyle Desrosiers, December 20, 2019, via telephone call between San Francisco, CA and Waco, TX, transcript, Baylor University Institute for Oral History, Waco. Interview transcript page 20.

[102]　Jesús Guillen, interview by Kyle Desrosiers, December 20, 2019, via telephone call between San Francisco, CA and Waco, TX, transcript, Baylor University Institute for Oral History, Waco. Interview transcript page 27.

Another participant, Ron, who was much younger and did not live through the AIDS crisis as an adult in the 1980s and 1990s, discussed the evolution of what he argues was once called the "gay community" into the "queer community". This evolution occurred as the way society understood gender and sexual identities was explored and expanded, and as the AIDS/HIV crisis forced LGBTQ folks to unite for common causes. Today, queer activism has only just begun to support and tentatively include transgender and nonbinary identities, and racial and ethnic minorities in addition to its historic representation of mainly gays and lesbians. In this way, some queers became assimilated in the mainstream society faster than the other queers.[103] About this, Ron expressed the shortcomings that he sees even within the LGBTQ community that others find liberating. As many others have, Ron has developed an ethic of accepting the good with the bad, but not settling for a community culture that is less inclusive, diverse, and informed than it could be. He learned what it meant to be stigmatized for being gay, Black, and HIV-positive. Ron's life situation lit a fire in his heart for justice, and because of that, he became an activist and advocate in the nonprofit sector. Due to his experiences with marginalization in society and in church, Ron questions all forms of fundamentalism: whether they are religious, social, or philosophical.

> I think that it's important that queer people really start to expand upon those things so that we don't fall into the trap of what so many people in religion do; you're not allowed to question certain things. You're not allowed to feel certain ways. You're not allowed to process information in certain ways and lead with what you think. You're supposed to go with this because you identify—I'm supposed to think, I'm supposed to go a certain way, I'm supposed to eat a certain places, shop at certain places, wear certain brands; I'm supposed to look a certain way because I identify as a gay man, and that, to me, is like, 'No. Why?' and it's because someone felt strong enough at some point to ask questions, and then, present themselves as they saw fit. [During the AIDS crisis] men, women, in-between, all the way above saw their friends just perish and got to a point where they became numb, and so, now that we live in an age of technology and science and U=U is a thing[104], traumatic experiences shaped the ideas in their approach to the subject, and so, it's hard for me to present a new perspective that is less traumatic that is going to trump that traumatic experience and teach a new perspective.[105]

The Reverend Charley explained that as a Protestant minister he has seen the way in which church culture and American norms have contributed to a sex-negative, body-negative culture. Rather than moving against the changing culture and shunning conversations that are scary or even threatening to long-held status quos, Charley argues that church can be the first place of change:

> I think some churches are only willing to go so far in the conversation and no further, and so, how it would take place—I think, if we go back to the true meaning of the word 'repentance': repentance [is] a changing of direction and if we can really embrace that. I think the other thing is the churches need to stop waiting on everybody feeling comfortable before they make the move. The whole thing about stepping out of faith, doing what you know is right even if it's uncomfortable, and then, moving into that comfort zone—if you're waiting for

---

[103] See: (Yep and Elia 2012; Byrd 2017; Pullen 2012).

[104] U=U stands for "undetectable means untransmittable." This means that people living with HIV, who maintain an undetectable viral load by taking an antiretroviral therapy drug, cannot sexually transmit the virus to others. Information obtained from National Institutes of Health (NIH). See (Courtenay-Quirk et al. 2006) for information about contemporary HIV stigma within the gay community, particular in reference to sex and relationships.

[105] Ron Wilson, interview by Kyle Desrosiers, September 19, 2019, in Dallas, Texas, transcript, Baylor University Institute for Oral History, Waco. From interview transcript page 14.

everybody to feel all nice, and warm, and fuzzy about it, and then come out with your statement affirming LGBT, it ain't never going to happen.[106]

Overwhelmingly, participants shared future perspectives that left room for both realism and optimism about the future. Much of the technology, legal protections, and community that exist today for HIV-positive and LGBTQ Americans has dramatically changed health outcomes, assimilation, and life quality. Yet, so many challenges lie ahead. In conclusion of this examination of the future, Robb commented that he no longer has religious faith in the traditional sense, but is hopeful about the eventual triumph of good:

It's hard not to be really angry right now at the destruction of our biosphere and, really, the representation in our government . . . [but] there is hope. I realize there is a lot of good going on far more than we realize; far, far more than is represented [in media]. There is a lot going on that's good in technology and mobilization of effort and resources. It is there. I really don't think we can grasp what could be done that is—it could happen overnight. I do believe that. There is hope.[107]

## 10. Conclusions: The Spiritual Narratives of Long-Term HIV Survivors

*When you fall from a great height, there is only one possible place to land: on the ground; the ground of truth . . . This was true of Gesar, the great warrior king of Tibet, whose escapades form the greatest epic of Tibetan literature. Gesar means 'indomitable,' someone who can never be put down. From the moment Gesar was born, his evil uncle Trotung tried all kinds of means to kill him. But with each attempt Gesar only grew stronger and stronger. It was thanks to Trotung's efforts that Gesar was to become so great. This gave rise to a Tibetan proverb: Trotung tro ma tung na, Gesar ge mi sar, which means that if Trotung had never been so malicious and scheming, Gesar could never have risen so high.*

—*The Tibetan Book of Living and Dying*, Songyal Rinpoche.[108]

*10.1. A Summary of Spiritual Narratives*

The experiences and beliefs of HIV-positive participants reveal a great depth of contemplation and spiritual inquiry. Participants in this project shared narratives of marginalization and mortality. They also reported dealing with alienation, seeking to understand paradox, and coping with death and suffering. These are all intrinsically spiritual/philosophical experiences. As professed by queer theorists and LGBTQ liberation theologists[109]—-some of whom identify still with Christian theology, and some of whom do not adhere closely to orthodoxy at all—there is special insight on freedom of thought that can be found by those who have been told they do not "belong". The corpus of these oral history interviews—both those featured and those not featured but preserved at the Baylor Institute for Oral History[110]—testifies to this argument.

Indeed, as the Tibetan parable of *Gesar* (above) explains, the religious institutions and doctrines that can cause harm—whether from hatred, apathy, or misplaced piety—*can* paradoxically act as a stimulus for deep spiritual inquiry in their victims. In liminal spaces of existential uncertainty, there is

---

[106] Charley Garrison, interview by Kyle Desrosiers, October 29, 2019, in Waco, Texas, transcript, Baylor University Institute for Oral History, Waco. Interview transcript page 17.

[107] Robb Ivey, interview by Kyle Desrosiers, September 20, 2019, Dallas, TX, transcript, Baylor University Institute for Oral History, Waco. Interview transcript page 24–25.

[108] (Rinpoche 2020, p. 36).

[109] See page 3 for more information on queer liberation theology.

[110] To access the corpus of audio recordings and transcripts for Kyle Desrosiers's interviews, please visit the Baylor Institute for Oral History digital collection at https://digitalcollections-baylor.quartexcollections.com/special-libraries-collections/oral-history.

a unique potential to challenge the status quo. This experience is reflected in the lives of many human beings who live with HIV/AIDS.

In the exploration of lifespan worldview development among people living with HIV, several key themes emerged:

First, *The Beginnings: Childhood Formation within Family and Faith Institutions* (Section 3) reports the ways in which participants first began to contemplate sexuality and identity in relationship to religious-cultural contexts. For example, some participants totally rejected the framework they inherited, for its homophobia or dogmatism. Others found that the framework they inherited—though contributing to alienation and trauma—also had components of goodness that were preserved into adulthood (which is explored more in later sections). While Jesús bemoaned the conditions of homophobia and rigidity ingrained in mid-twentieth century Roman Catholic Christianity and Mexican morality, he also recognized the positive role that faith played in shaping people who are loving and compassionate, such as his grandmother. Ron endured conversion therapy in the Church of Jesus Christ of Latter-day Saints as a young man, which he reported destroyed his self-esteem and hope. Michael reported feeling alienated from his peers for being gay and Jewish in a town with few members of either communities. Negative and abusive experiences in youth, as well as dogmatic frameworks, left participants lacking answers to their deepest questions–and worse, led to feelings of isolation and lasting trauma. Later, however, participants report changes and continuities with the systems they inherited.

Next, *Understanding Sexuality: Discovery, Liberation, and Trauma* (Section 4) reports narratives of early responses to religious and cultural paradigms of "normative" sexuality and identity. Participants—who identified as gay, queer, or LGBTQ—reported experiences whereby they rejected or reexamined messages about their "sanctity", "sinfulness", or unbelonging. For example, Ron, a Black gay man explains that growing up poor in the inner city meant that religion was one of the few constants in his community life, which made it only more difficult to abandon. Robb explains that meeting Japanese people on his Mormon mission helped him see the beauty and goodness of other worldviews he was taught to fear. Michael told about giving his parents an ultimatum: to love and accept him and his partner Don, or lose him forever; poignantly, he says he is glad to have done this and that his parents eventually came around, noting: "If I hadn't disclosed that to them, how much love I would've cheated myself out of."[111]

*The Onset of HIV: Personal and Collective Memories* (Section 5) focused on narratives from the early days of the epidemic, when participants first recognized the threat of HIV/AIDS to their community and confronted their own positive diagnoses. These memories represent a significant turning point in the spiritual journeys of respondents: men who were young and healthy had to suddenly cope with a terrifying new health crisis. Michael remembered sitting with his gay friend group in a West Hollywood café in 1981, unable to believe what they read about the "gay epidemic" in the newspaper; out of all those men, he said he is the only one from that night who is still alive. Ron remembered being paralyzed by fear when he learned of his HIV-positive status and feeling guilty that he might have infected his partner, whom he found out later had HIV before him and lied about his status, making Ron feel unfairly guilty. Jesús summed up the feeling in the gay community in those days: "'Man, what is worse: really be[ing] with a community and see all your friends and lovers die, or to know that you might die next day and have nobody to tell that you might die?'"[112]

Section 6.1. *Answering Tough Questions* and Section 6.2. *Cultivating New Spiritual Practices* collected narratives of how spirituality was used to understand the HIV/AIDS crisis and participant's fear of imminent mortality. Participants frequently reported that their worldviews expanded as they strove

---

[111]　Michael Sugar, interview by Kyle Desrosiers, October 2, 2019, telephone interview between Los Angeles, CA and Waco, TX, transcript, Baylor University Institute for Oral History, Waco, TX. Interview transcript page 26.

[112]　Jesús Guillen, interview by Kyle Desrosiers, December 20, 2019, via telephone call between San Francisco, CA and Waco, TX, transcript, Baylor University Institute for Oral History, Waco.

to cope with mortality, marginalization, and loss. Robb's experiences on a Mormon mission and at BYU helped him later on in life to cope with his status. Similarly, Michael's reconciliation and bonding with his father over the HIV/AIDS crisis helped lead him back to his faith. Many participants reported having developed new spiritual practices or participated in new spiritual experiences that brought them feelings of peace, well-being, and liberation. For Robb and Jesús, this included transcendental psychological experiences like meditation, prayer, seeing visions, and using learning as a spiritual practice. For Ron, who is an atheist, liberation was an intellectual endeavor found through study. Jesús reconnected with the shamanism of his indigenous Mexican roots. Michael commented on the joy and peace his rabbi, a lesbian activist in San Francisco, helped him discover in his Jewish spiritual life. Most participants reported an increase in compassion for other humans and the environment.

Section 7.1. *Coping with Collective Trauma* and Section 7.2. *Dealing with Personal Trauma* reported beliefs and experiences that were used to navigate collective trauma and suffering, as well as experiences with personal traumatic experiences. These experiences ranged from fearing mortality in the wake of their diagnoses, to understanding how to cope with suffering and death among those whom they knew and loved. Additionally, participants had to cope with the societal marginalization that came with being HIV-positive. These experiences put many of the participants in a position, they reported, where it became necessary to answer tough questions of meaning and purpose, and how to seek peace during chaos. Many participants reported the struggle that they faced knowing they survived while others with HIV did not. Other participants needed answers to the question of "why" such horror could happen and sought more comprehensive paradigms leaving more room for nuance and complexity than those they were taught.

Common to many interviews, such as with Michael and Ron, was an anger at the federal government for its inaction, as well as an anger for the leadership of organized religion for both their inaction and distinct efforts to stifle HIV care/research and support for gay and trans people living with HIV. Many participants reported that their friends were dying; for two decades the older participants endured an era without effective HIV medication, where AIDS was often a death sentence. However, sources of peace and comfort for both personal and collective trauma were healthy spiritual institutions such as Reform synagogues and the Metropolitan Community Church, romantic partners, family, social activism, friendships, films and art, pets, and learning.

In Section 7.2. *Dealing with Personal Trauma* participants shared examples of unhealthy romantic relationship and intracommunal violence or abuse. For example, Robb shared a story of how the man who spread the HIV virus to him intentionally hid his status and had unprotected sex with Robb for the sole purpose of infecting him. Jesús feared the disclosure of his HIV status, as he was a Mexican citizen and knew he could be deported—so he had a friend perform the blood test in his place. Ron saw a link between the experience of Blackness in America and marginalization for being LGBTQ and HIV-positive; compounded identities produced stigma and alienation both external and internal to the gay community.

Next, Section 8. *The Spiritual Project of Dealing with Loss and Death* discussed the relationship between spirituality and mourning. Michael mentioned the significance of the American AIDS Quilt as a means to process the loss of hundreds of thousands of HIV-positive Americans, dozens of whom he knew personally. In this section, it was also very common for participants to report the difference that affirming clergy had made in their lives as marginalized people who were coping with loss. A Protestant minister, the Rev. Charley, spoke of the need for a compassionate, humble pastoral ethic of accompaniment for HIV ministry. Ron, Michael, and Jesús found healing and meaning through service and activism with nonprofits such as ACTUP. Activism combined the desire to enjoin with collective quasi-spiritual causes greater than one's self, and also gain a sense of honoring the memory and life of the deceased. Ultimately, mourning defines the story of HIV and AIDS, but mourning is not the end of the story; there is resistance, persistent dignity in the faith of apathy, activism, and above all, relentless love. Long-term HIV survivors have frequently confronted more loss than many Americans, and it has played a deep role in their religious/worldview journeys.

Finally, Section 9. *Hopes and Anxieties about the Future: from Spiritual Awakening to Ethical Healthcare* reported how survivors view their personal and collective futures. They shared beliefs and opinions about the fate of the HIV-positive and LGBTQ communities domestically and abroad. Some of the older participants such as Robb expressed his mourning for a gay community that was once stronger, tighter-knit, and more united than he says it is today. Jesús expressed fear and frustration of the younger generation not taking the HIV virus seriously, viewing it as "gay diabetes" thanks to their Western privilege, where it is incorrectly perceived to be easily treatable. Almost universally, participants expressed that because *they* as HIV-positive individuals and queer people have endured such marginalization, they are also especially attuned to the oppression of others, such as refugees and migrants at the southern border. Ultimately, participants voiced the importance of hope—not a blind hope that trusts in dogma—but a hope that is made consummate through actions of peace and justice.

After examining trends common to the lifespan experiences of HIV-positive participants in this project, it is likely that HIV/AIDS status has been the impetus for major spiritual, worldview, and philosophical investigation for many people. Being HIV-positive engenders experiences that intimately relate to health, sexuality, mortality, societal expectations, legal marginalization, and ethics. It is a condition that has necessitated a deep spiritual and philosophical inquiry for many survivors. Those offering medical or pastoral care should not view HIV survivors as lucky, nor as persons to be pitied. These survivors have narratives of loss and hardship, but also stories of bravery, compassion, and service. This collection of narratives provides a cross-section of the histories of HIV-positive men and their spiritual life. Hopefully, these narratives contribute to the discourse on human rights, ethics, and faith as they relate to the HIV virus. Indeed, it is from the marginalized—such as those with chronic illness and those close to mortality—that one can learn much about the nature of faith.

*10.2. Limitations and Objectives for Future Research*

There were a few points of limitation in this project. First, this author would have liked a diverse participant pool, which he will prioritize as he continues to interview HIV-positive Americans about religion, belief, and spirituality. While participants came from an array of ethnic identities, there was low representation of women, and no transgender or nonbinary participants. Additionally, the author was not able to recruit or speak with anyone from outside Jewish and Christian faith origins, nor Asian Americans. Much time was spent contacting HIV/AIDS institutions and organizations, but due to the sensitive and stigmatized nature of the content, recruitment proved difficult. Additionally, because many contemporary Christian and Jewish sects are far more open to LGBTQ affirmation and have a more progressive stance on HIV/AIDS than many American Muslim or Hindu communities, for example, the author—an undergraduate student with limited credentials—was only able to obtain interviews with the former. Thus, future research needs to include religions outside of Christianity, Judaism, and secularism, especially Islam, Hinduism, Buddhism, and Sikhism, and others that play significant roles in the United States.

The author hopes to personally interview a more religiously diverse demographic for further contribution to the HIV and Spirituality Archive at the Baylor Institute for Oral History, and also recommends that other researchers also seek to expand these inquiries of HIV and religion to include these diverse demographics.

**11. Materials and Methods**

The objective of this project was to collect, organize, and present a body of qualitative data about the longitudinal lived experiences and evolving beliefs of long-term survivors living with HIV/AIDS in America. The author of this article, Kyle Desrosiers, conducted a combination of in-person and remote telephone interviews with 10 participants in Texas, California, Louisiana, and Oklahoma. This research

was facilitated by the Baylor Institute for Oral History[113], and it is thanks to the institute that this author accessed recording and transcribing technology, as well as a permanent location to archive the corpus of these interviews.

Following interviews, the audio recordings were generously transcribed by the undergraduate and graduate student team at the Baylor University Institute for Oral History. Next, the author began to analyze the text transcripts for themes. Lastly, themes were organized, and quotes were offered as qualitative data for each theme, to show continuity, discontinuity, similarity, and difference among the participants who reported their experiences with religion, spirituality, culture, and HIV. This work was originally conducted for an undergraduate thesis in the Baylor University Honor College University Scholars program. From the corpus of the entire archival work, 5 participants have been selected to show an array of diverse queer men's perspectives on long-term HIV survivorship. The finished project offers a glimpse into the beliefs and experiences of individuals among a group of people, focused on a particular theme—spiritual lifespan development—which is underrepresented in existing works of oral history profiling HIV/AIDS and/or LGBTQ persons.

The driving research questions behind the interviews were: (1) what is the relationship of members of the HIV-positive community in America to religion and spirituality?; (2) what experiences shaped changes or continuity in belief and practice?; and (3) how have worldview changes over time, as experienced by HIV survivors related to their interactions with family, religion, and community?

Certain resources were helpful in the development of a qualitative data analysis method. First, the 2013 sourcebook *Qualitative Data Analysis* by Miles and Huberman[114] was invaluable. Second, the 2011 *Oxford Handbook of Oral History*[115] provided guidance in the development of an oral history project, and aided with methods of collection and presentation. Lastly, the previous work by the author's faculty advisor, Dr. Mia Moody-Ramirez, particularly in her and Dr. C. Burleson's work *Sixteen Ain't So Sweet: Longitudinal Study of Jasper Dragging* (2014)[116], was inspirational and instructional in better understanding how to present oral history, especially when a tragic and traumatic history must be told with dignity, such as was the case in both the history of the Jasper lynching of a Black man and the suffering experienced by HIV-positive Americans.

The audio and transcript interviews can be accessed online at the Institute for Oral History archive, but due to the long-term scope of this process and extensive time required to compile such an archive, and the slowed workflow thanks to the novel 2020 Coronavirus pandemic in the world, these resources may not be accessible in the public domain until several months after this publication. Please contact the author for more information. All materials will be available upon request at the Baylor Oral History Institute and can be accessed digitally or at the office in the Texas Collection at Baylor University, Waco, TX.

**Supplementary Materials:** The interviews cited in this article are housed in the database of the Baylor Institute for Oral History, Baylor University, Waco, Texas. To access the corpus of audio recordings and transcripts for Kyle Desrosiers's interviews, please visit the Baylor Institute for Oral History digital collection at https://digitalcollections-baylor.quartexcollections.com/special-libraries-collections/oral-history.

**Funding:** This research received no external funding.

**Acknowledgments:** I offer my sincerest thanks to the interview participants for this project. May their stories continue to educate us and call us to action. I also offer special thanks to the esteemed faculty of the Baylor Institute for Oral History, including especially Stephen Sloan, Dianne Reyes, and the student workers who helped transcribe the interviews used in this corpus. I am grateful to my mentor and thesis advisor, Mia Moody-Ramirez,

---

[113] The Baylor University Institute for Oral History, located in Waco, Texas, is a freestanding research department within Baylor University's Division of Academic Affairs. Please visit https://www.baylor.edu/oralhistory/ for more information, or contact the office at +1(254)-710-3437. Not all archives may be publicly available due to the labor intensive archival process and the ongoing global COVID-19 health crisis. All materials will be available upon request.

[114] (Miles and Huberman 1994).

[115] (Ritchie 2010).

[116] (Moody-Ramirez and Burleson 2014).

whose work on Black media representation inspired my own on work on queer and religious identities and representation. Lastly, I would like to thank another close mentor of mine, Jennifer Good, who offered advice and support in the research process.

**Conflicts of Interest:** The author declares no conflict of interest.

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
