# Peer review of "Spiritual Reports from Long-Term HIV Survivors: Reclaiming Meaning While Confronting Mortality"

_religions, doi:10.3390/rel11110602_

Round 1
Reviewer 1 Report
It should be noted at the outset that the submitted paper was not anonymized (see part No. 11 at the end of the text) ...! How do the editors react to such a transgression?
From the overall point of view, the article is written at a high professional level, understandable, clear, the findings are well analyzed, categorized, and summarized in the conclusion. The author builds on very difficult, precise, beneficial qualitative research, which will be useful also for other researchers, for which he deserves praise and respect.
Partial formal comments include:
- he identifies the Mormons (Latter-day Saints) with Christianity, which is impermissible in terms of the definition of Christianity; Christians consider the Mormons to be an eccentric sect and have nothing to do with them;
- in note No. 102 is perhaps a typing error: drug / virus (transmit the drug to others);
- paragraph 1388 - 1394 is, in my opinion, superfluous, written in an unacademic style, rather journalistic, it addresses the reader, which does not belong to academic writing.
Other comments have a philosophical-theological character and may serve the author's future research and publication:
- He raises the question of the contrast between human rights and religion (line 123n). This question deserves a deeper reflection: what are human rights, what are their origins, how does theology approach them (specifically the Christian theology, if we criticize the relationship of Christianity to human rights) ... What is superior (human rights to religion or vice versa)? What does this mean for the concept of supreme authority (is it the Christian God, and if not, then who / what ...)? This also has consequences for the issue of clinging to the status quo, which the author criticizes (lines 1214 - 1215). Is the status quo bad and progress, in any case, good and desirable? Should the church be first in line for change?
- When criticizing Christians' attitudes toward LGBTQ people, it is important to examine, to what extent it is the failure of specific ecclesial communities and leaders, and to what extent the often-mentioned "hatred" is contained in the "essence" of the religion. In his authoritative books, documents, interpretations ... Is there ...?
- The red thread of the article is the contrast between religion and non-institutionalized spirituality (547-549, 1270-1272, etc.), while personal spirituality not tied to a formal religious institution is perceived as better, more holistic, less fundamental (547-549), deeper, healthier (623), radical freedom of thought (1266). The term nonreligious antifundamentalism is also mentioned (1195). This would require a comparison with theological works devoted to the problem of freedom, mysticism, the inner spiritual "castle" - and God's place in it.
In conclusion, I note that the text is interesting, but very partial, which is directly related to its qualitative nature. Qualitative research of this type should be supplemented by a broader theological, philosophical, anthropological basis + quantitative research of a similar nature, so that the results are more comprehensive. I therefore leave it up to the editors to assess whether they also accept such partial contributions to the professional discussion.
Author Response
Dear reviewer,
Thank you so much for your time and effort in this review process. I really appreciate your thoroughness, as I am striving to make this article as successful as possible. I will respond to your comments line-by-line, and I will attach a revised version of the document, if you would like to look at changes I have made.
It should be noted at the outset that the submitted paper was not anonymized (see part No. 11 at the end of the text) ...! How do the editors react to such a transgression?
—––Thank you for pointing this out. It is a transgression, and one I apologize for. I am a graduate student who is excited for the opportunity to publish my very first contribution, so I would ask for patience. I have worked hard and will continue to make amendments as the article reviewers request.
From the overall point of view, the article is written at a high professional level, understandable, clear, the findings are well analyzed, categorized, and summarized in the conclusion. The author builds on very difficult, precise, beneficial qualitative research, which will be useful also for other researchers, for which he deserves praise and respect.
Partial formal comments include:
- he identifies the Mormons (Latter-day Saints) with Christianity, which is impermissible in terms of the definition of Christianity; Christians consider the Mormons to be an eccentric sect and have nothing to do with them;
—––Regarding Mormons (Latter-day Saints), I have chosen to represent the religion and the individual who was raised in it as he requested to be represented. There are thousands of sects that consider themselves to be “Christian”, and indeed there is vicious debate and prejudice that exists about many of these sects towards other sects. We can see this in anticatholicism and antiprotestantism, and arguments about salvation, right belief, etc. In order to avoid partiality, I cannot make a claim that my personal definition of what Christianity is should govern the paper. Whether I believe Mormons to be “Christian” or not, the aim of this paper is not a theological argument, but rather to present information from oral history interviews, qualitative findings from the mouths of interviewees directly. Latter-day Saints identify around the teaching of Jesus, and revere the Christian scriptures. Though the LDS Church professes continuing revelation and other authorities since the Bible, it nonetheless finds its roots in the Christian tradition. Wherever possible I simply used “Roman Catholic”, “Mormon/LDS”, “Metropolitan Community Church” to identify very specific denominations of different faith traditions, rather than “Christian” to avoid such a controversy.
- in note No. 102 is perhaps a typing error: drug / virus (transmit the drug to others);
—––Thank you for catching this. I changed it.
- paragraph 1388 - 1394 is, in my opinion, superfluous, written in an unacademic style, rather journalistic, it addresses the reader, which does not belong to academic writing.
—––In the updated manuscript on the MDPI page, there is a section bracketed by lines 1387-1400. I believe that it is to this that you are referring. I have made some changes in an effort to rewrite this conclusion paragraph in a more objective way. Please review the reframing and let me know your opinion. That would be appreciated. Lines 1387-1398 in the latest edition after edits.
Other comments have a philosophical-theological character and may serve the author's future research and publication:
- He raises the question of the contrast between human rights and religion (line 123n). This question deserves a deeper reflection: what are human rights, what are their origins, how does theology approach them (specifically the Christian theology, if we criticize the relationship of Christianity to human rights) ... What is superior (human rights to religion or vice versa)? What does this mean for the concept of supreme authority (is it the Christian God, and if not, then who / what ...)? This also has consequences for the issue of clinging to the status quo, which the author criticizes (lines 1214 - 1215). Is the status quo bad and progress, in any case, good and desirable? Should the church be first in line for change?
—––These are great comments. I am interested in ethics and theology, and would indeed like to examine these topics in future research. Also, I am going to scan through the paper, and look for sections that might be “too journalistic” or relating to opinions, and try to revise.
- When criticizing Christians' attitudes toward LGBTQ people, it is important to examine, to what extent it is the failure of specific ecclesial communities and leaders, and to what extent the often-mentioned "hatred" is contained in the "essence" of the religion. In his authoritative books, documents, interpretations ... Is there ...?
—––I tried hard to represent specific narratives of individuals’ experiences, without generalizations. I am going to go back and make sure wording makes this clear. The failure to act quickly enough to save lives in the case of AIDS has been connected by writers such as Randy Schilts, who blames specific leaders in the evangelical churches, the “New Right” supported by televangelists, certain doctrines in certain churches, certain concepts of “morality”, certain opinions of laypeople of various levels of religious education who believe their faith teaches specific things, whether or not it does/ or intrinsically has to. I will reexamine wording. The object is not to slander Christianity in a partial way. For example, see where Charlie (MCC pastor) says:
About the relationship between religious stigma and HIV/AIDS, a Waco, Texas pastor, the Reverend Charley, who is not HIV-positive, but offers pastoral care to many HIV-positive survivors at his Metropolitan Community Church parish through support groups and a food pantry, voiced profound hurt and disgust at the way the Christian gospel has in some instances been used to justify negligence, violence, and apathy by those in power. Due to this, Charley offers a compassionate Christian response to those who can no longer trust Christianity, or any religious tradition because of the harm it caused them and those they love(d):
All we knew was our friends were dying, and so, there was a lot of questioning, and certainly, when you’re harangued by religious fundamentalists who are saying, ‘It’s because God is killing you and for good reason, ‘yes, there will be people who abandon religion, and I can understand why that would be the case. That’s still the case today, not necessarily because of HIV, but there are still religious fundamentalists who oppress the LGBT community and because of that oppression, they walk away from any form of organized—especially Christian—religion, and all I can say is, ‘I understand.’ (lines 720-734)
- The red thread of the article is the contrast between religion and non-institutionalized spirituality (547-549, 1270-1272, etc.), while personal spirituality not tied to a formal religious institution is perceived as better, more holistic, less fundamental (547-549), deeper, healthier (623), radical freedom of thought (1266). The term nonreligious antifundamentalism is also mentioned (1195). This would require a comparison with theological works devoted to the problem of freedom, mysticism, the inner spiritual "castle" - and God's place in it.
—––This would be a fascinating topic to continue to pursue. Do you think any texts relating to these topics are needed in this article, or do you only mean to suggest that it would be good for future research?
In conclusion, I note that the text is interesting, but very partial, which is directly related to its qualitative nature. Qualitative research of this type should be supplemented by a broader theological, philosophical, anthropological basis + quantitative research of a similar nature, so that the results are more comprehensive. I therefore leave it up to the editors to assess whether they also accept such partial contributions to the professional discussion.
—––Thank you for putting so much time and thought into your comments and review. I am taking these comments seriously as I strive to put forth the best paper possible.
All the best.

Reviewer 2 Report
I found a few very minor editing issues and one word use that raised a question in my mind:
Line 8: Something missing. Perhaps, “comes from?”
Line 5:should be “the study of epidemiology”
62: Is histography the correct term?
This is an excellent study of a topic--the AIDS crisis--that needs more analysis. The author is absolutely correct that the study of religion is usually undertaken as a bookish academic pursuit that seems divorced from the lived experiences of actual people. This study seeks to address that, presenting heart-breaking yet beautiful stories of people of a variety of different faith backgrounds who lived through an event made far more horrible than it needed to be, if the society and government had handled it more humanely.
Kudos to the author. I look forward to seeing this in a published format. I teach a couple of classes I may use it in, and I will share it with my colleagues.
Author Response
Dear reviewer,
Thank you for your time and support. I appreciate the grammatical and word usage edits. I made the changes in line 8 (line 12 on my manuscript), and the line 5 (55 on my manuscript, so perhaps this is a typo). Thanks for noticing these mistakes.
Also, yes, histography was the wrong word. I changed it to "creative nonfiction work", which is an accurate description.
All the best.